# Language-driven Scene Synthesis using Multi-conditional Diffusion Model

**An Dinh Vuong**
FSOFT AI Center
Vietnam

**Minh Nhat Vu**
TU Wien, AIT GmbH
Austria

**Toan Tien Nguyen**
FSOFT AI Center
Vietnam

**Baoru Huang**
Imperial College London
UK

**Dzung Nguyen**
FSOFT AI Center
Vietnam

**Thieu Vo**
Ton Duc Thang University
Vietnam

**Anh Nguyen**
University of Liverpool
UK

## Abstract

Scene synthesis is a challenging problem with several industrial applications. Recently, substantial efforts have been directed to synthesize the scene using human motions, room layouts, or spatial graphs as the input. However, few studies have addressed this problem from multiple modalities, especially combining text prompts. In this paper, we propose a language-driven scene synthesis task, which is a new task that integrates text prompts, human motion, and existing objects for scene synthesis. Unlike other single-condition synthesis tasks, our problem involves multiple conditions and requires a strategy for processing and encoding them into a unified space. To address the challenge, we present a multi-conditional diffusion model, which differs from the implicit unification approach of other diffusion literature by explicitly predicting the guiding points for the original data distribution. We demonstrate that our approach is theoretically supportive. The intensive experiment results illustrate that our method outperforms state-of-the-art benchmarks and enables natural scene editing applications. The source code and dataset can be accessed at `https://lang-scene-synth.github.io/`.

## 1 Introduction

Scene synthesis has gained significant attention from the research community in the past few years (Ye et al., 2022). This area of research has numerous applications such as virtual simulation, video animation, and human-robot communication (Yi et al., 2023). Recently, many scene synthesis methods have been proposed (Ye et al., 2022; Yi et al., 2023; Wang et al., 2019a). However, most of these methods generate objects based on furniture distribution (Paschalidou et al., 2021) rather than customized user preferences. On the contrary, arranging household appliances is a personalized task, in which the most suitable room layout is usually the one that fits our eyes best (Chang et al., 2017). In this work, we hypothesize that the scene can be synthesized from multiple conditions such as human motion, object location, and customized user preferences (*i.e.*, text prompts). We show that the text prompt input plays an important role in generating physically realistic and semantically meaningful scenes while enabling real-world scene editing applications.

Existing literature on scene synthesis often makes assumptions about scene representation, such as knowing floor plan (Wang et al., 2018) or objects are represented using bounding boxes (Paschalidou

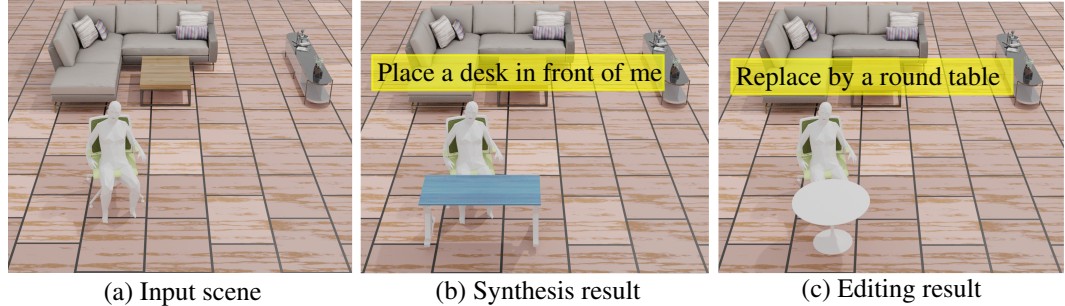

| (a) Input scene | (b) Synthesis result | (c) Editing result |

Figure 1: We introduce *Language-driven Scene Synthesis* task, which involves the leverage of human-input text prompts to generate physically plausible and semantically reasonable objects.

et al., 2021). These assumptions limit the ability to represent the object in a finer manner or edit the scene. To address these limitations, we revisit the 3D point clouds representation (Zhao et al., 2021), a traditional approach that can capture both the spatial position and the shape of objects (Rusu et al., 2007). The use of 3D point clouds can represent objects that are not directly linked to human movement, which is a constraint in the representation of contact points in (Ye et al., 2022), and lead to a more comprehensive and meticulous representation of the scenes.

To synthesize the scene from multiple conditions, including existing objects of the scene represented by 3D point clouds, human motion, and their textual command, we propose a new multi-conditional diffusion model. Although several multi-conditional diffusion models have been presented recently (Nichol et al., 2021), most of them focus on images/videos and employ *implicit* unification of all latent spaces (Ma et al., 2023; Nguyen et al., 2023; Qin et al., 2023; Zhang and Agrawala, 2023; Zhao et al., 2023; Ruan et al., 2022). On the other hand, we introduce the first multi-conditional diffusion model on 3D point clouds and an *explicit* mechanism to combine multiple conditions. Specifically, we introduce a concept of guiding points for scene synthesis and theoretically show that the guiding points explicitly contribute to the denoising process. The intensive empirical evidence confirms our findings.

Taking advantage of using the text prompt input, we introduce a new family of editing operations for the scene synthesis task, including object replacement, shape alternation, and object displacement. Inspired by (Tseng et al., 2022; Tevet et al., 2022; Song et al., 2020; Saharia et al., 2022), our proposed method enables users to select an object they want to modify and specify their desired changes via textual command. Our experimental results show that the proposed methodology yields encouraging editing results, demonstrating the potential of our method to bridge the gap between research and practical real-world applications. Our contribution can be summarized as:

- We present the language-driven scene synthesis task, a new challenge that generates objects based on human motions and given objects while following user linguistic commands.

- We propose a new multi-conditional diffusion model to tackle the language-driven scene synthesis task from multiple conditions.

- We validate our method empirically and theoretically, and introduce several scene-editing applications. The results show remarkable improvements over state-of-the-art approaches.

## 2 Related Works

**Scene Representation.** Generating new objects in a scene requires a scene representation method. Prior works have proposed many approaches to address this problem. Fisher et al. (2012) projects objects onto the 2D surface and sample jittered points for each object. Similarly, 2D heatmaps are implemented in (Qi et al., 2018). However, the major limitation of both mentioned approaches is that boundaries of 2D projections often do not capture sufficient details of 3D objects (Yi et al., 2019). Recent approaches, such as ATISS (Paschalidou et al., 2021) and MIME (Yi et al., 2023) represent each object as a 3D bounding box using a quadruple of category, size, orientation, and position, which has limitation in estimating the corresponding shape of the object. Additionally, Ye et al. (2022) represent objects as contact points with a sequence of human movements, but this approach

only works for objects attached to human motions and ignores detached objects. To overcome these shortcomings, we propose a representation method based on 3D point clouds (Qi et al., 2017a) that estimates not only the positions of room objects but also their shapes (Yi et al., 2019).

**Scene Synthesis.** The literature about scene synthesis can be categorized into the following groups. *i) Scene synthesis from floor plan* (Paschalidou et al., 2021; Wang et al., 2019a, 2021; Ritchie et al., 2019; Wang et al., 2018): generating objects that fit a given floor layout, which clearly lacks incorporating human interactions with objects. *ii) Scene synthesis from text* (Chang et al., 2015, 2014, 2017; Savva et al., 2017; Ma et al., 2018): completing a scene given a textual description of the room arrangement. The majority of these works do not consider human motions in the generated scenes. The prevalent solution, for instance, SceneSeer (Chang et al., 2017) implements a scene template capturing the relationship between room objects. However, this method necessitates considerable manual effort, resulting in inflexibility. *iii) Scene synthesis from human motions* (Yi et al., 2023; Ye et al., 2022; Nie et al., 2022; Yi et al., 2023): generating objects aligning with a human motions sequence. The assumption in this direction is more practical since humans are interactive, but the existing issue is that the generated object often complies with the dataset distribution than personalized intentions. *iv) Scene synthesis from spatial graph* (Jiang et al., 2018; Li et al., 2019; Dhamo et al., 2021; Wang et al., 2019a; Qi et al., 2018): rendering scene from a spatial graph. Despite having a great potential to describe large 3D scenes (Wald et al., 2020), spatial graphs require regular updates to account for entities such as humans with multifaceted roles, which results in extensive computation (Gadre et al., 2022). Since humans usually arrange objects with their personalized intentions, we propose a task that comprises both text prompts and human motions as the input, which aims to overcome the domain gap from scene synthesis research to real-life practice.

**Diffusion Models for Scene Synthesis.** Diffusion probabilistic models are a group of latent variable models that leverage Markov chains to map the noise distribution to the original data distribution (Sohl-Dickstein et al., 2015; Song and Ermon, 2020; Ho et al., 2020; Luo and Hu, 2021). In Dhariwal and Nichol (2021), classifier-guided diffusion is introduced for conditional generation tasks and has since been improved by Nichol et al. (2021); Ho et al. (2022). Subsequently, inspired by these impressive results, numerous works based on conditional diffusion models have achieved state-of-the-art outcomes in different generative tasks (Liu et al., 2023; Tevet et al., 2022; Tseng et al., 2022; Le et al., 2023; Huang et al., 2023). However, in our problem settings, the condition is a complicated system that includes not only textual commands but also the existing objects of the scene and human motions. Therefore, a solution to address multi-conditional guidance is required. While multi-conditional diffusion models have been actively researched, most of them focus on images/videos and utilize implicit unification of all latent spaces (Ma et al., 2023; Ruan et al., 2022; Zhang and Agrawala, 2023; Zhao et al., 2023). On the other hand, we propose a novel approach to handle multi-conditional settings by predicting the guiding points as the mean of the original data distribution. We demonstrate that the guiding points explicitly contribute to the synthesis results.

## 3 Methodology

### 3.1 Problem Formulation

We input a text prompt $e$ and a partial of the scene $\mathcal{S}$, which consists of human pose $H$ and a set of objects $\{O_1, O_2, \ldots, O_M\}$. Each object is represented as a point cloud: $O_i = \{(x_0^i, y_0^i, z_0^i), \ldots, (x_N^i, y_N^i, z_N^i)\}$. Let $\mathbf{y} = \{e, H, O_1, \ldots, O_M\}$ be the generative conditions. Our goal is to generate object $O_{M+1}$ consistent with the room arrangement of $\mathcal{S}$, human motion $H$, and semantically related to $e$:

$$\max \Pr\left(O_{M+1} | \mathbf{y}\right) . \tag{1}$$

We design a guidance diffusion model to handle the task. The forward process is adapted as in (Ho et al., 2020), and our primary focus is on developing an effective backward (denoising) process. Our problem involves multiple conditions, so we propose fusing all conditions into a new concept called *Guiding Points*, which serves as a comprehensive feature representation. We provide theoretical evidence demonstrating that guiding points explicitly contribute to the backward process, unlike the previous implicit unification of multi-conditional diffusion models in the literature (Sohl-Dickstein et al., 2015; Song and Ermon, 2020; Ho et al., 2020; Luo and Hu, 2021).

## 3.2 Multi-conditional Diffusion for Scene Synthesis

**Forward Process.** Given a point cloud $\mathbf{x}_0$ drawn from the interior of $O_{M+1}$. We add Gaussian noise each timestep to obtain a sequence of noisy point clouds $\mathbf{x}_1, \mathbf{x}_2, \ldots, \mathbf{x}_T$ as follows:

$$q(\mathbf{x}_{t+1}|\mathbf{x}_t) = \mathcal{N}(\sqrt{1-\beta_t}\mathbf{x}_t, \beta_t\mathbf{I}); \quad q(\mathbf{x}_{1:T}|\mathbf{x}_0) = \prod_{t=0}^{T-1} q(\mathbf{x}_{t+1}|\mathbf{x}_t), \tag{2}$$

where variance $\beta_t$ is determined by a known scheduler. Eventually when $T \to \infty$, $\mathbf{x}_T$ is equivalent to an isotropic Gaussian distribution.

**Backward Process.** The goal of the conditional reverse noising process is to determine the backward probability $\hat{q}(\mathbf{x}_t|\mathbf{x}_{t+1}, \mathbf{y})$. We begin with the following proposition:

**Proposition 1.**

$$\hat{q}(\mathbf{x}_t|\mathbf{x}_{t+1}, \mathbf{y}) = \frac{q(\mathbf{x}_t|\mathbf{x}_{t+1})\hat{q}(\mathbf{y}|\mathbf{x}_t)}{\hat{q}(\mathbf{y}, \mathbf{x}_{t+1})} \mathbb{E}\left[q(\mathbf{x}_{t+1}|\mathbf{x}_0)\right]. \tag{3}$$

*Proof.* See Appendix. $\square$

*Remark* 1.1. The expression in Eq. (3) provides a more explicit formulation of $\hat{q}(\mathbf{x}_t|\mathbf{x}_{t+1}, \mathbf{y})$ in comparison to the conditional backward process presented in (Dhariwal and Nichol, 2021). This enables a further examination of the impact of the multi-condition $\mathbf{y}$ on the prediction of $\mathbf{x}_0$.

*Remark* 1.2. Sohl-Dickstein et al. (2015) indicate that generating canonical samples from Eq. (3) is often infeasible. Assume $\mathbf{x}_0$ is drawn from a uniform distribution over a domain of $\mathbf{S}$ and $q(\mathbf{y}|\mathbf{x}_0)$ is non-zero uniform over $\mathbf{S}$, we discretize the expression in Eq. (3) by

$$\hat{q}(\mathbf{x}_t|\mathbf{x}_{t+1}, \mathbf{y}) \approx \frac{q(\mathbf{x}_t|\mathbf{x}_{t+1})\hat{q}(\mathbf{y}|\mathbf{x}_t)}{\hat{q}(\mathbf{y}, \mathbf{x}_{t+1})} \frac{1}{|\hat{\mathbf{S}}|} \sum_{\mathbf{x}_0 \in \hat{\mathbf{S}}} q(\mathbf{x}_{t+1}|\mathbf{x}_0)q(\mathbf{x}_0), \tag{4}$$

with $\hat{\mathbf{S}}$ being the sampling set of $\mathbf{x}_0$. Using Bayes' theorem, we obtain:

$$q(\mathbf{x}_0) = \frac{q(\mathbf{x}_0)q(\mathbf{x}_0, \mathbf{y})}{q(\mathbf{x}_0, \mathbf{y})} = \frac{q(\mathbf{x}_0, \mathbf{y})}{q(\mathbf{y}|\mathbf{x}_0)} = \frac{q(\mathbf{x}_0|\mathbf{y})q(\mathbf{y})}{q(\mathbf{y}|\mathbf{x}_0)}. \tag{5}$$

As $q(\mathbf{y})$ is independent with denoising process, $q(\mathbf{x}_0)$ is uniform, and $q(\mathbf{y}|\mathbf{x}_0)$ is non-zero uniform over $\mathbf{S}$; we infer $q(\mathbf{x}_0|\mathbf{y})$ is also uniform over $\mathbf{S}$. In addition, $q(\mathbf{x}_0) \neq 0 \Leftrightarrow q(\mathbf{x}_0|\mathbf{y}) \neq 0$; thus, $\hat{\mathbf{S}}$ can also be considered as a sampling set of $\mathbf{x}_0|\mathbf{y}$. We can now derive Eq. (4) using Eq. (5) as follows:

$$\hat{q}(\mathbf{x}_t|\mathbf{x}_{t+1}, \mathbf{y}) \approx \kappa q(\mathbf{x}_t|\mathbf{x}_{t+1})\hat{q}(\mathbf{y}|\mathbf{x}_t) \frac{1}{|\hat{\mathbf{S}}|} \sum_{\mathbf{x}_0|\mathbf{y} \in \hat{\mathbf{S}}} q(\mathbf{x}_{t+1}|\mathbf{x}_0)q(\mathbf{x}_0|\mathbf{y}), \tag{6}$$

where $\kappa = q(\mathbf{y})/(q(\mathbf{y}|\mathbf{x}_0)\hat{q}(\mathbf{y}, \mathbf{x}_{t+1}))$. Denote $\mu_0$ as the mean of the initial probability distribution $q(\mathbf{x}_0)$. Consequently, $\mu_0|\mathbf{y}$ can be used to approximate $\mathbf{x}_0|\mathbf{y}$. We further consider $\tilde{\mathbf{S}}$, the sampling set of estimated points of $\mu_0$ as an approximation of $\hat{\mathbf{S}}$; therefore, Eq. (6) can be reduced to:

$$\hat{q}(\mathbf{x}_t|\mathbf{x}_{t+1}, \mathbf{y}) \approx \kappa q(\mathbf{x}_t|\mathbf{x}_{t+1})\hat{q}(\mathbf{y}|\mathbf{x}_t) \frac{1}{|\tilde{\mathbf{S}}|} \sum_{\mathbf{x}_0|\mathbf{y} \in \tilde{\mathbf{S}}} q(\mathbf{x}_{t+1}|\mathbf{x}_0)q(\mathbf{x}_0|\mathbf{y}). \tag{7}$$

In our problem settings, $\mathbf{x}_0$ is drawn from the interior of an object uniformly; therefore, Remark 1.2 is applicable. Inspired by Tevet et al. (2022), we model the conditional reverse noising process in Eq. (7) using the following reduction form:

$$\hat{q}(\mathbf{x}_t|\mathbf{x}_{t+1}, \mathbf{y}) \approx f(\mathbf{x}_{t+1}, \mathbf{y}, \tilde{\mathbf{S}}), \tag{8}$$

where $f$ is a neural network. The main difference between our proposal and related works is that we incorporate the term guiding points $\tilde{\mathbf{S}}$ into the backward probability.

### 3.3 Guiding Points Network

**Observation.** The text-based modality is a reliable and solid source of information to synthesize unseen objects, given partial entities of the scene (Chang et al., 2014). Indeed, while many works contemplate floor plan as a significant input (Wang et al., 2019a; Yi et al., 2023; Paschalidou et al., 2021), we find that the floor plan is redundant for generating the target object when the context is known. Consider the example in Fig. 2a- 2c, which is also common in not only our problem but also in our daily life. In this scene, a human is sitting on a chair with another nearby; suppose the human gives a command: "*Place a desk to the right of me and next to the chair.*". Regardless of the floor plan, there is only an optimal setting location, depicted in Fig. 2b, for the desk. Therefore, the text-based modality provides enough information to achieve our task.

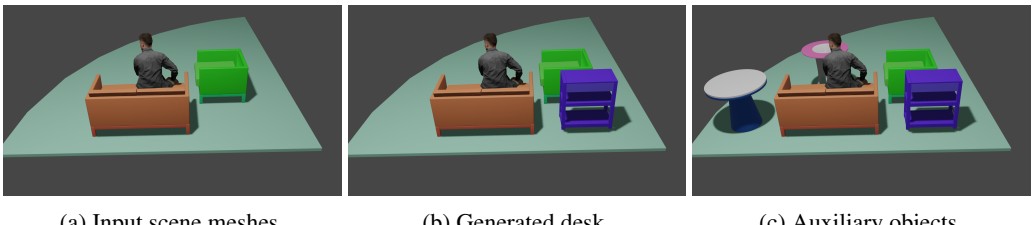

(a) Input scene meshes.      (b) Generated desk.      (c) Auxiliary objects.

Figure 2: **Motivational example.** In this example, we want to generate an object aligned with the following command: "*Place a desk to the right of me and next to the chair.*"

In this example, what is the most crucial factor that the modal command offers? The answer is the *spatial relation* implied from the linguistic meanings, which is also indicated in Chang et al. (2014). We further argue this spatial relationship is entity-centric as it only prioritizes certain scene entities. Indeed, if there are two round tables added to the scene as in Fig. 2c, the optimal location for the target object remains the same. Consequently, the targeted object primarily depends on the position of the human and the chair, *i.e.*, a defined set of scene entities.

To align each given scene entity with the target object $O_{M+1}$, we use 3D transformations approach (Besl and McKay, 1992). Specifically, for a given point cloud $O_i = \{o_0^i, o_1^i, \ldots, o_N^i\}$, our objective is to compute a transformation matrix for each point $o_j^i$ given by:

$$\mathbf{F}_{i;j} = \begin{bmatrix} \alpha_x^i & \beta_x^i & \gamma_x^i & t_x^i \\ \alpha_y^i & \beta_y^i & \gamma_y^i & t_y^i \\ \alpha_z^i & \beta_z^i & \gamma_z^i & t_z^i \\ 0 & 0 & 0 & 1 \end{bmatrix}. \tag{9}$$

We can predict each point $\hat{o}_i$ of the target object by applying transformation matrix $\mathbf{F}_{i;j}$ to each $o_j^i$:

$$\begin{bmatrix} \hat{x}_j^i \\ \hat{y}_j^i \\ \hat{z}_j^i \\ 1 \end{bmatrix} = \begin{bmatrix} \alpha_x^i & \beta_x^i & \gamma_x^i & t_x^i \\ \alpha_y^i & \beta_y^i & \gamma_y^i & t_y^i \\ \alpha_z^i & \beta_z^i & \gamma_z^i & t_z^i \\ 0 & 0 & 0 & 1 \end{bmatrix} \begin{bmatrix} x_j^i \\ y_j^i \\ z_j^i \\ 1 \end{bmatrix}, \tag{10}$$

where $o_j^i = (x_j^i, y_j^i, z_j^i)$ and $\hat{o}_j^i = (\hat{x}_j^i, \hat{y}_j^i, \hat{z}_j^i)$. We refer this concept of predicting $\hat{o}_j^i$ as guiding points, which has been introduced in the main paper.

Next, as observed in Fig. 2c, we find that each object $O_i$ contributes differently in providing clues to locate the object $O_{M+1}$. This implies that each object should possess its own attention score, represented by a scalar $\mathbf{w}_i$, which resembles the famous attention technique (Vaswani et al., 2017). We employ this attention mechanism to determine attention matrix $\mathbf{w} = [\mathbf{w}_0, \mathbf{w}_1, \ldots, \mathbf{w}_M]$ of the set of objects and utilize the 3D transformation in Eq. (10) to determine guiding points.

**Guiding Points Prediction.** Fig. 3 illustrates the method overview. We hypothesize that the target object can be determined from the scene entities using 3D transformation techniques (Park et al., 2017). Each scene entity contributes a distinct weight to the generation of the target object, which can be inferred from the spatial context of the text prompt as discussed in our Observation. Our

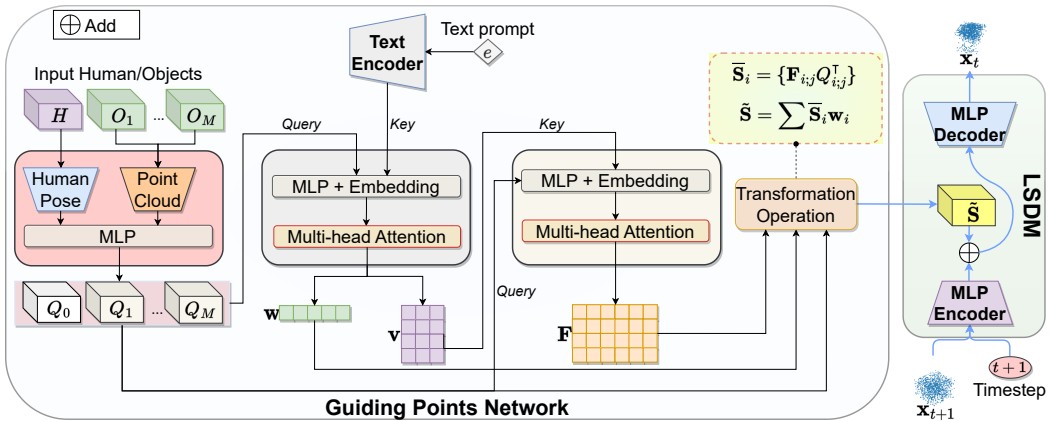

Figure 3: **Methodology overview.** Our main contribution is the *Guiding Points Network*, where we integrate all information from the conditions $\mathbf{y}$ to generate guiding points $\tilde{\mathbf{S}}$.

central component, the Guiding Points Network utilizes the theory in Section 3.2 to establish a translation vector between each scene entity and the target object and then utilizes the high-level translation to determine point-wise transformation matrices for each scene entity. Finally, guiding points are obtained by aggregating the contributions of all scene entities, which are weighted by their corresponding attention scores.

Initially, we extract point-level features from the input by feeding human motions $H$ into a human pose backbone (Hassan et al., 2021) and objects $O_1, O_2, \ldots, O_M$ into a point cloud backbone (Qi et al., 2017a) to obtain point-level representations: $[Q_0, Q_1, Q_2, \ldots, Q_M]$. We encode $e$ using a text encoder (Radford et al., 2021) to obtain the text feature $e'$. Following the point-level extraction for each scene entity, we extract spatial information from the text prompt by utilizing the off-the-shelf multi-head attention layer (Vaswani et al., 2017), where the input key is the text embedding $e'$, the input queries are the given scene entities. The attention layer computes two outputs: $\mathbf{w}$, which represents the weights that indicate the contribution of each scene entity to the generation of guiding points, and $\mathbf{v}$, which denotes the translation vectors between the scene entities and the target object.

We then utilize the weight $\mathbf{w}$ and translation vectors $\mathbf{v}$ to calculate the guiding points $\tilde{\mathbf{S}}$ using 3D transformations. From the $\mathbf{v}_i$ of each scene entity, we predict a set of transformation matrices $\mathbf{F}_i$ for each point of that scene entity. This approach, apart from applying an identical translation vector to every point, diversifies the point set distribution, therefore, can capture the shape of the target object. The guiding points $\overline{\mathbf{S}}_i$ derived from $Q_i$ are then calculated by applying the corresponding point-wise transformation matrices $\mathbf{F}_i$ to the points of the scene entity (Fig. 3). Finally, we composite all relative guiding points from each entity with its corresponding $\mathbf{w}_i$ as follows: $\tilde{\mathbf{S}} = \sum_{i=0}^{M} \overline{\mathbf{S}}_i \mathbf{w}_i$.

**Output Procedure.** Eventually, we want to sample $\mathbf{x}_t$ given the input $\mathbf{x}_{t+1}$ and conditions $\mathbf{y}$. We compute the signal $\mathbf{x}_t$ given by: $\mathbf{x}_t = \text{MLP}(\mathbf{x}_{t+1}, t+1, \tilde{\mathbf{S}})$. More information about the model and training procedure can be found in the Appendix.

## 4  Experiment

We first compare our method for the scene synthesis task with other recent work and demonstrate its editing applications. We then assess the effectiveness of the guiding points to the synthesis results. All experiments are trained on an NVIDIA GeForce 3090 Ti with 1000 epochs within two days.

### 4.1  Language-driven Scene Synthesis

**Datasets.** We use the following datasets to evaluate the language-driven scene synthesis task: *i) PRO-teXt:* We contribute PRO-teXt, an extension of PROXD (Hassan et al., 2019) and PROXE (Zhang et al., 2020). There are 180/20 interactions for training/testing in our PRO-teXt dataset. *ii) HUMANISE:*

We utilize 143/17 interactions of HUMANISE (Wang et al., 2022) to train/test. Details of the datasets are specified in the Appendix.

**Baselines.** We compare our approach **L**anguage-driven **S**cene Synthesis using Multi-conditional **D**iffusion **M**odel (LSDM) with: *i) ATISS* (Paschalidou et al., 2021). An autoregressive model that outputs the next object given some objects of the scene. *ii) SUMMON* (Ye et al., 2022). A transformer-based approach that predicts the human's contact points with objects. *iii) MIME* (Yi et al., 2023). An autoregressive model based on transformer techniques to handle human motions. Since the language-driven scene synthesis is a new task and none of the baselines utilize the text prompt, we also set up our model without using the text input for a fair comparison. *iv) MIME + text embedding.* We extend MIME with a text encoder to handle the text prompts; the represented text features are directly concatenated to the transformer layers of the original MIME's architecture. *v) MCDM.* We implement a multi-conditional diffusion model (MCDM) that directly combines all features from the input (3D point clouds, text prompts).

**Metrics.** As in (Lyu et al., 2021), we employ three widely used metrics in 3D point cloud tasks, Chamfer distance (CD), Earth Mover's distance (EMD), and F1 score, to evaluate the degree of correspondence between the predicted point cloud and the ground truth of the target object.

| | PRO-teXt | | | HUMANISE | | |
|---|---|---|---|---|---|---|
| Baseline | CD ↓ | EMD ↓ | F1 ↑ | CD ↓ | EMD ↓ | F1 ↑ |
| ATISS (Paschalidou et al., 2021) | 2.0756 | 1.4140 | 0.0663 | 5.3595 | 2.0843 | 0.0308 |
| SUMMON (Ye et al., 2022) | 2.1437 | 1.3994 | 0.0673 | 5.3260 | 2.0827 | 0.0305 |
| MIME (Yi et al., 2023) | 2.0493 | 1.3832 | 0.0990 | 5.4259 | 2.0837 | 0.0628 |
| MIME (Yi et al., 2023) + text embedding | 1.8424 | 1.2865 | 0.1032 | 4.7035 | 1.8201 | 0.0849 |
| MCDM | 0.6301 | 0.7269 | 0.3574 | 0.8586 | 0.8757 | 0.2515 |
| LSDM w.o. text (Ours) | 0.9134 | 1.0156 | 0.0506 | 1.1740 | 1.1128 | 0.0412 |
| LSDM (Ours) | **0.5365** | **0.5906** | **0.5160** | **0.7379** | **0.7505** | **0.4395** |

Table 1: **Scene synthesis results.** Bold and underline are the best and second-best, respectively.

**Quantitative Results.** Table 1 summarises the results of our method and other approaches. In the absence of text prompts, our method shows remarkable improvements in CD and EMD metrics, while MIME achieves the highest F1 score. When text prompts are employed, our LSDM yields dominant results compared to all baselines in all metrics. Notably, the F1 scores of our method are approximately $1.45$ times and $1.76$ times greater than the second-best in PRO-teXt and HUMANISE, respectively. This result confirms the importance of text prompts and our explicit strategy for the scene synthesis task.

**Qualitative Results.** We utilize the object recovery algorithm (Ye et al., 2022) and 3D-FUTURE (Fu et al., 2021) to synthesize 3D scenes. Fig. 4 demonstrates the qualitative results. It is clear that our proposed method successfully generates more suitable objects, while other benchmark models fail to capture the desired positions. More qualitative results can be found in our demonstration video.

**User Study.** We ask 40 users to evaluate the generated results of all methods from the PRO-teXt and HUMANISE datasets. 8 output videos are generated from each method and placed side-by-side. The participants evaluate based on three criteria: naturalness, non-collision, and intentional matching. Fig. 5 shows that our method is preferred over the compared baselines in most cases and is the closest to the ground truth.

## 4.2 Scene Editing with Text Prompts

The use of text prompts provides a natural way to edit the scene. We introduce three operations based on users' text prompts: *i) Object Replacement*: given an object $O_{M+1}$, the goal is to replace $O_{M+1}$ with a new object $O_{M+1}^*$ via a text prompt $e^*$. *ii) Shape Alternation*: we alter the shape of an object $O_{M+1}$ by a text prompt $e^*$ to obtain a new shape built upon $O_{M+1}$. We follow Tevet et al. (2022) by retaining a quarter of $O_{M+1}$'s point cloud, which contains the lowest $z$-coordinate points (i.e., those closest to the ground) while diffusing the remaining 75% through our pre-trained model. *iii) Object Displacement*: we want to move object $O_{M+1}$ to a new position using a textual command $e^*$.

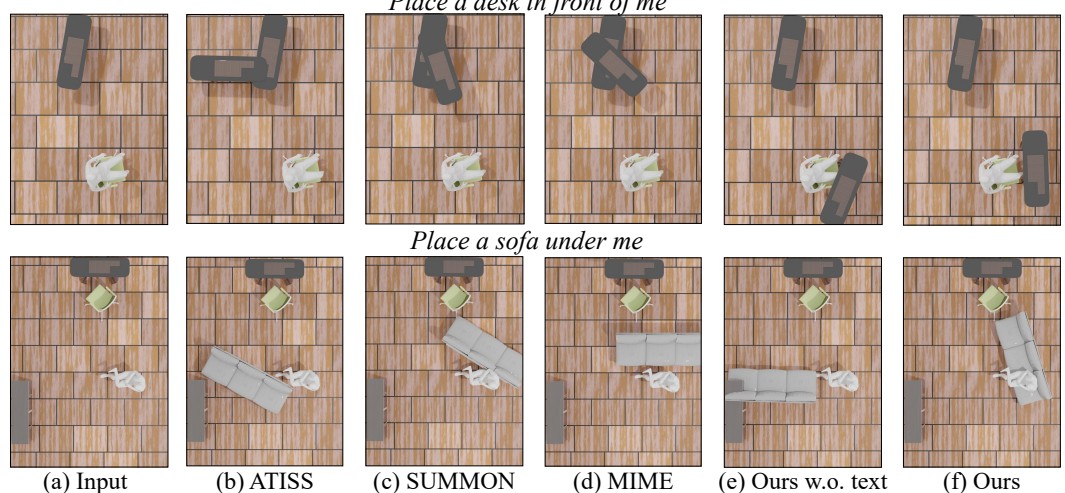

*Place a desk in front of me*

*Place a sofa under me*

| (a) Input | (b) ATISS | (c) SUMMON | (d) MIME | (e) Ours w.o. text | (f) Ours |

Figure 4: **Scene visualization.** We present illustrative examples generated by all baseline methods on two test cases: generating a desk (first row) and a sofa (second row).

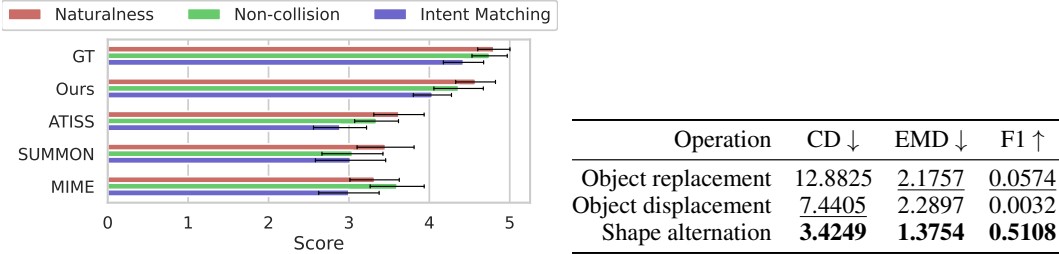

Figure 5: **User evaluation of all baselines.**

| Operation | CD ↓ | EMD ↓ | F1 ↑ |
|---|---|---|---|
| Object replacement | 12.8825 | 2.1757 | 0.0574 |
| Object displacement | 7.4405 | 2.2897 | 0.0032 |
| Shape alternation | **3.4249** | **1.3754** | **0.5108** |

Table 2: **Scene editing results.**

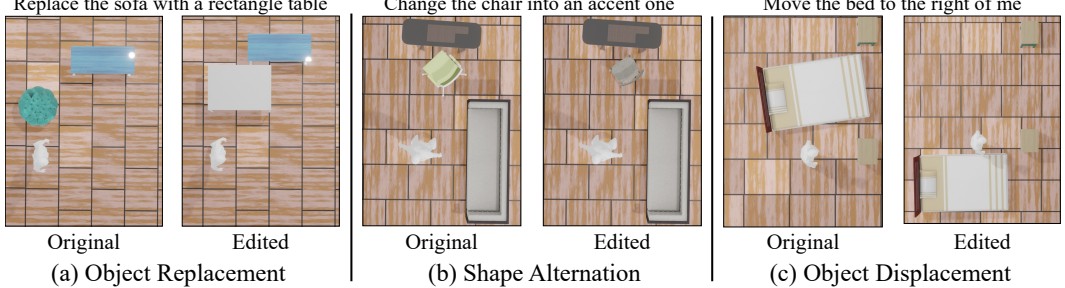

| Replace the sofa with a rectangle table | Change the chair into an accent one | Move the bed to the right of me |
| Original          Edited | Original          Edited | Original          Edited |
| (a) Object Replacement | (b) Shape Alternation | (c) Object Displacement |

Figure 6: **Scene editing visualization.** We provide qualitative results of three introduced editing operations. The text prompts are indicated above each example.

**Qualitative Results.** Fig. 6 illustrates the proposed editing operations, showcasing the promising and semantically meaningful outcomes obtained by our method. The above settings are close to real life, which bridges the gap between the scene synthesis literature and practical applications.

**Quantitative Results.** Table 2 shows the quantitative results of scene editing operations. The experiment setups are described in our Appendix. Table 2 demonstrates that our approach yields decent performance for the shape alternation operation while there is still scope for improvement in the other operations. It is worth noting that shape alternation is relatively easier than object replacement and displacement as inherits 25% of the target object as the input.

## 4.3 Ablation Study

This section presents an in-depth study of the proposed neural architecture. We show the impact of each modality on the overall results. We also provide a detailed analysis of the importance and effectiveness guiding points technique.

**How does each modality contribute to the performance?** To understand the contributions of each component utilized to predict guiding points as well as overall results, we conduct a comprehensive analysis in Table 3. In the first row, our model does not predict the high-level translation vector $\mathbf{v}$, which results in the absence of guiding points. When our model does not predict $\mathbf{F}$ (second row), it utilizes $\mathbf{v}$ to generate an identical transformation matrix for each point of an entity. In the next two rows, only a partial representation of the scene (human/objects) is used. Table 3 shows that the utilization of $\mathbf{v}, \mathbf{F}$, as well as the inclusion of text prompts, humans, and existing objects substantially impact the prediction of guiding points and the final results.

| Baseline | Input used | $\tilde{\mathbf{S}}$ | CD ↓ | EMD ↓ | F1 ↑ |
|---|---|---|---|---|---|
| LSDM w.o. predicting $\mathbf{v}$ | ∅ | none | 4.6172 | 2.1086 | 0.0391 |
| LSDM w.o. predicting $\mathbf{F}$ | text, human, objects | partial | 1.8933 | 1.1350 | 0.2400 |
| LSDM predicting $\tilde{\mathbf{S}}$ from only objects | text, objects | partial | 1.5050 | 1.0653 | 0.3185 |
| LSDM predicting $\tilde{\mathbf{S}}$ from only human | text, human | partial | 1.0119 | 0.8419 | 0.3855 |
| LSDM (ours) | text, human, objects | full | **0.5365** | **0.5906** | **0.5160** |

Table 3: **Components Analysis.** We assess the performance of our method under different settings.

**Can guiding points represent the target object?** Recall that the guiding points are estimations of the mean $\mu_0$. To determine whether our method can reasonably predict $\mu_0$ from the conditions $\mathbf{y}$, we provide both quantitative and qualitative assessments. We first compare $s^2$ and $d_0^2$ in Table 4. The relatively small errors suggest that our model has good generalization capability towards unseen objects, as the forecast guiding points are in close proximity to the ground truth centroids. In addition, we provide demonstrations of our proposed guiding points mechanism. Fig. 7 depicts guiding points $\tilde{\mathbf{S}}$ in three test scenarios. The visualization illustrates that our guiding points (red) extend seamlessly over the centroids of the target objects (blue). This outcome fulfills our original intent when synthesizing the new object based on the guiding points. Furthermore, referring to Table 3 and 4, we observe that as the MSE decreases, the model's performance improves, providing empirical confirmation of the usefulness of guiding points.

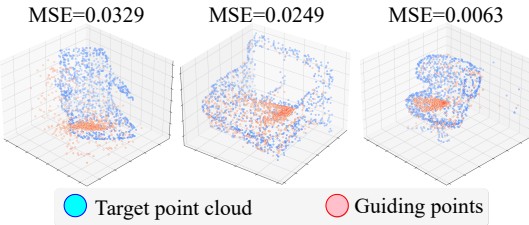

MSE=0.0329    MSE=0.0249    MSE=0.0063

○ Target point cloud    ○ Guiding points

| Baseline | MSE ↓ |
|---|---|
| LSDM w.o. predicting $\mathbf{F}$ | 0.5992 |
| LSDM predicting $\tilde{\mathbf{S}}$ from only objects | 0.4618 |
| LSDM predicting $\tilde{\mathbf{S}}$ from only human | 0.3388 |
| LSDM (ours) | 0.2091 |
| Minimal squared distance $d_0^2$ | **0.0914** |

Figure 7: **Guiding points visualization.** The MSE is indicated above each sample.

Table 4: **Guiding points evaluation.** We calculate the MSE between predicted $\tilde{\mathbf{S}}$ and the centroids of target objects.

**How does each scene entity contribute to the generation of guiding points?** Fig. 8a demonstrates how the attention mechanism determines the most relevant objects providing information for the generation of guiding points. In this example, in response to the text prompt "Place a one-seater sofa under me and beside the sofa", the utilized mechanism assigns the highest attention weight to the human and subsequently to the sofa numbered 2 referenced in the prompt. This is consistent with the semantic meaning of the text prompt, which enables our model to correctly leverage the attention weights from scene entities to generate guiding points.

**How many guiding points are needed for the estimation process?** Fig. 8b demonstrates the influence of the number of guiding points used in the estimation process on the CD and F1 metrics.

The result illustrates that the effectiveness of our method is directly proportional to the number of guiding points employed. However, the performance of our approach reaches a plateau as the number of guiding points surpasses 1024. Consequently, to optimize the results as well as computational complexity, we set the number of guiding points to 1024.

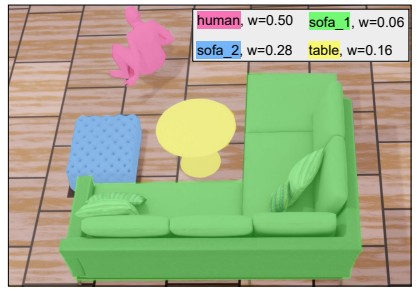

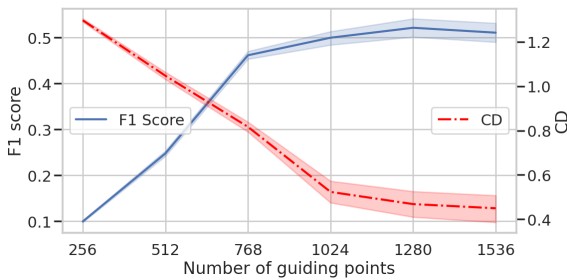

(a) Normalized attention weights of scene entities.

(b) Impact of the number of guiding points.

Figure 8: **Additional guiding points analysis.** We present: (a) how our model attends to scene entities to determine guiding points and (b) a study on scaling guiding points.

## 5 Discussion

**Limitations.** Although achieving satisfactory results and meaningful scene editing applications, our LSDM approach still holds some limitations. First, our theoretical findings have an assumption constrained to uniform data like point clouds. Second, we train the guiding point network jointly with the denoising process; therefore, the predicted guiding points are not always aligned with the target object. Some failure cases of guiding points are visualized in the Appendix. In addition, our scene editing demonstrates modest results and necessitates future improvements.

**Broader Impact.** We believe our paper has two key applications. First, our proposed language-driven scene synthesis task can be applied to metaverse and animation. For instance, a user enters an empty apartment and gives different commands (e.g., "*Putting a sofa in the corner*," "*Placing a table next to the sofa*") to arrange the room furniture (where physical contact is not mandatory). Second, our proposed guiding points concept is a general concept and can be applied beyond the scope of scene synthesis literature. Guiding points can be applied to visual grounding tasks such as language driven-grasping (Vuong et al., 2023), where we can predict *guiding pixels* indicating possible boxes from the conditions to guide the denoising process.

## 6 Conclusion

We have introduced language-driven scene synthesis, a new task that incorporates human intention into the process of synthesizing objects. To address the challenge of multiple conditions, we have proposed a novel solution inspired by the conditional diffusion model and guiding points to effectively generate target objects. The intensive experimental results show that our method significantly improves over other state-of-the-art approaches. Furthermore, we have introduced three scene editing operations that can be useful in real-world applications. Our source code, dataset, and trained models will be released for further study.

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

## A    Remarks on Related Work

**Human-aware Scene Synthesis.** Although there are many scene synthesis works, a few of them consider human motions. To our best knowledge, MIME (Yi et al., 2023) and Ye et al. (2022) are the most related literature to ours. We compare our paper with other works in Table 5.

|  | Scene Representation | Text input | Customized | Editable |
|---|---|---|---|---|
| SUMMON (Ye et al., 2022) | Contact points | ✗ | ✗ | ✗ |
| MIME (Yi et al., 2023) | 3D bounding box | ✗ | ✗ | ✗ |
| Ours | Point cloud | ✓ | ✓ | ✓ |

Table 5: **Our work vs. related human-aware scene synthesis findings.**

We further remark on the metrics utilized in our study in comparison to other related works. In (Ye et al., 2022), two metrics, namely reconstruction accuracy and consistency score, are employed to evaluate the predicted contact labels against the ground truth, primarily focusing on the reconstruction of room objects. However, since our approach directly predicts the point cloud of the target object, metrics that specifically assess the reconstruction of 3D point clouds, such as CD, EMD, and F1, are more suitable for addressing the original objective outlined in (Ye et al., 2022).

Yi et al. (2023) adopt three metrics: interpenetration, IoU, and FID score. Although FID is a solid metric, we contend that the FID score is not adequate for our proposed task, as a single orthogonal top-down view may not capture sufficient details of 3D scenes (Li et al., 2022). Lastly, the IoU metric can be well-covered by the metrics we have employed in this study, namely CD, EMD, and F1. The metric of interpenetration is expanded in our 3D configuration in the following manner:

$$3\text{D IP} = \frac{\sum_{i=1}^{M} \sum_{p \in \hat{O}_{M+1}} \mathbb{I}_{p \in O_i} + \sum_{p \in \hat{O}_{M+1}} \mathbb{I}_{p \in H}}{|\hat{O}_{M+1}|}, \tag{11}$$

where we denote $\hat{O}_{M+1}$ as the target object predicted by any baseline, and $\mathbb{I}$ indicates whether a point $p$ belonging to the predicted point cloud lies within the interior of the scene entity or not. Supplementary results on FID and IP metrics are shown in Appendix Sec. F.

**Point Cloud Estimation.** PointNet (Qi et al., 2017a) stands as one of the pioneering deep learning approaches for extracting point cloud data representations. Followed by this seminal work, a variety of methodologies have been utilized to tackle the problem of estimating point sets (Zhou et al., 2021): GNN-based (Wang et al., 2019b), GAN-based (Achlioptas et al., 2018; Cai et al., 2020; Li et al., 2021), flow-based (Yang et al., 2019), transformer-based (Zhao et al., 2021), etc. Overall, prior works have shown remarkable results on 3D deep learning tasks, such as semantic/part segmentation and shape classification (Zhao et al., 2021). Since designing point cloud neural networks that have the ability to create photorealistic, novel, and distinctive shapes remains a challenge (Li et al., 2021); therefore, in this paper, we do not establish a novel point cloud network but rather use existing baselines to evaluate the scene synthesis problem.

## B    Conditional Denoising Process with Guiding Points

Following the *Conditional Reverse Noising Process* (Dhariwal and Nichol, 2021), we use the similar definition of the conditional noising $\hat{q}$, given by:

$$\hat{q}(\mathbf{x}_0) \stackrel{\text{def}}{=} q(\mathbf{x}_0), \tag{12}$$

$$\hat{q}(\mathbf{x}_{t+1}|\mathbf{x}_t, \mathbf{y}) \stackrel{\text{def}}{=} q(\mathbf{x}_{t+1}|\mathbf{x}_t), \tag{13}$$

$$q(\mathbf{y}|\mathbf{x}_0) = \text{Known per sample}, \tag{14}$$

$$\hat{q}(\mathbf{x}_{1:T}|\mathbf{x}_0, \mathbf{y}) \stackrel{\text{def}}{=} \prod_{t=0}^{T-1} \hat{q}(\mathbf{x}_{t+1}|\mathbf{x}_t, \mathbf{y}). \tag{15}$$

**Proof of Proposition 1.**

It follows from (Dhariwal and Nichol, 2021, pages 25-26) that

$$\hat{q}(\mathbf{x}_t) = q(\mathbf{x}_t) , \tag{16}$$

and

$$\hat{q}(\mathbf{x}_t|\mathbf{x}_{t+1}, \mathbf{y}) = \frac{q(\mathbf{x}_t|\mathbf{x}_{t+1})\hat{q}(\mathbf{y}|\mathbf{x}_t)}{\hat{q}(\mathbf{y}|\mathbf{x}_{t+1})} . \tag{17}$$

By using Bayes' Theorem and Eq. (16), we have

$$\hat{q}(\mathbf{y}|\mathbf{x}_{t+1}) = \frac{\hat{q}(\mathbf{y}, \mathbf{x}_{t+1})}{\hat{q}(\mathbf{x}_{t+1})} = \frac{\hat{q}(\mathbf{y}, \mathbf{x}_{t+1})}{q(\mathbf{x}_{t+1})} . \tag{18}$$

We can progressively compute $q(\mathbf{x}_{t+1})$ as follows

$$q(\mathbf{x}_{t+1}) = \int_{\mathbf{S}} q(\mathbf{x}_{t+1}, \mathbf{x}_0)\mathrm{d}\mathbf{x}_0 = \int_{\mathbf{S}} q(\mathbf{x}_{t+1}|\mathbf{x}_0)q(\mathbf{x}_0)\mathrm{d}\mathbf{x}_0 = \mathbb{E}\left[q(\mathbf{x}_{t+1}|\mathbf{x}_0)\right] . \tag{19}$$

Combining Eqs. (18) and (19) into Eq. (17), we can verify that

$$\hat{q}(\mathbf{x}_t|\mathbf{x}_{t+1}, \mathbf{y}) = \frac{q(\mathbf{x}_t|\mathbf{x}_{t+1})\hat{q}(\mathbf{y}|\mathbf{x}_t)}{\hat{q}(\mathbf{y}, \mathbf{x}_{t+1})}\mathbb{E}\left[q(\mathbf{x}_{t+1}|\mathbf{x}_0)\right] . \tag{20}$$

Thus, Proposition 1 is then proved.

**Proposition 2.** *Let $\mathbf{G}$ be the convex hull of $\mathbf{S}$ and $d_0$ be the minimum distance from any point on a facet of $\mathbf{G}$ to the centroid $\mu_0$. By denoting $\mathbf{G}^\circ$ as the interior of $\mathbf{G}$ and assuming $\tilde{\mathbf{S}} = \{r_1, r_2, \ldots, r_L\}$, $r_i \sim \mathcal{N}(\mu_0, \sigma_0^2\mathbf{I})$, and $s^2 = \frac{1}{L}\sum_{i=1}^{L}\|r_i - \mu_0\|^2 > \frac{C\sigma_0^2}{L}$, we show that*

$$\Pr\left(r_i \in \mathbf{G}^\circ\right) \geq \frac{1}{2}\left(1 + \mathrm{erf}\left(\frac{d_0}{O(s)}\right)\right) . \tag{21}$$

*Proof.* Since any point that has the distance to $\mu_0$ less than $d_0$ must lie within the interior of $\mathbf{G}$ (Boyd and Vandenberghe, 2004). Therefore

$$\Pr\left(r_i \in \mathbf{G}^\circ\right) \geq \Pr\left(\|r_i - \mu_0\| < d_0\right) . \tag{22}$$

It is sufficient to prove the Proposition 2 with an alternative term $\Pr\left(\|r_i - \mu_0\| < d_0\right)$. Since $r_i$ is drawn from $\mathcal{N}(\mu_0, \sigma_0^2\mathbf{I})$, we have that

$$\Pr\left(\|r_i - \mu_0\| < d_0\right) = \frac{1}{2}\left(1 + \mathrm{erf}\left(\frac{d_0}{\sigma_0\sqrt{2}}\right)\right) . \tag{23}$$

According to the definition of the standard deviation, the following is obtained

$$s^2 = \frac{1}{L}\sum_{i=1}^{L}\|r_i - \mu_0\|^2 . \tag{24}$$

Since each $r_i \sim \mathcal{N}(\mu_0, \sigma_0^2\mathbf{I})$, we can write $r_i = \mu_0 + \sigma_0\epsilon_i$, where $\epsilon_i \sim \mathcal{N}(0, \mathbf{I})$. The Eq. (24) can be rewritten as

$$s^2 = \frac{\sigma_0^2}{L}\sum_{i=1}^{L}\|\epsilon_i\|^2 . \tag{25}$$

Eq. (25) indicates that $\frac{s^2 L}{\sigma_0^2}$ follows a chi-squared distribution with $L$ degrees of freedom. Since $s^2 > \frac{C\sigma_0^2}{L}$, we infer that $\sigma_0 = O(s)$. Therefore, we can rewrite Eq. (23) as follows

$$\Pr\left(\|r_i - \mu_0\| < d_0\right) = \frac{1}{2}\left(1 + \mathrm{erf}\left(\frac{d_0}{O(s)}\right)\right) . \tag{26}$$

From Eqs. (22) and (26), we conclude this proposition. $\square$

The condition $s^2 > \frac{C\sigma_0^2}{L}$ is applicable, as we have proved in the Appendix that $\frac{s^2 L}{\sigma_0^2}$ follows a chi-squared distribution with $L$ degrees of freedom. By considering the specific value $C = 1.0$, we can easily check that the probability of $\frac{s^2 L}{\sigma_0^2} > C$ is greater than $1 - 10^{-9}$ when $L$ exceeds 20. We establish the following corollary:

**Corollary 2.1.**

$$\lim_{s \to 0} \Pr \left( r_i \in \mathbf{G}^\circ \right) = 1, \forall r_i \in \tilde{\mathbf{S}} \,. \tag{27}$$

Verification of Corollary 2.1 is straightforward, as $\mathrm{erf}(z)$ in Eq. (21) is known to be monotonically increasing to 1 when $z \to \infty$ (DeGroot and Schervish, 2012). Furthermore, this corollary implies that a smaller MSE ($s^2$) between the predicted guiding points $\tilde{\mathbf{S}}$ and $\mu_0$ corresponds to a more accurate sampling set. Our experiments (In Sec. 4.3) also validate this implication.

## C Implementation Details

Let $N$ be the number of points in each scene entity. We begin with conducting point-level feature extraction from the given scene arrangement by $Q_0 = \mathrm{HumanPoseBackbone}(H_0) \in \mathbb{R}^{N \times 3}$ and $[Q_1, Q_2, \ldots, Q_M] = \mathrm{PointCloudBackbone}\left([O_1, O_2, \ldots, O_M]\right) \in \mathbb{R}^{M \times N \times 3}$.

Subsequently, the input text prompt $e$ is embedded using a text encoder via: $\tilde{e} = \mathrm{TextEncoder}(e) \in \mathbb{R}^D$. The dimension of the text encoder backbone, denoted as $D$, varies depending on the specific architecture. In order to standardize the output of all text encoders, we apply linear layers as follows: $e' = \mathrm{LinearLayers}(\tilde{e}) \in \mathbb{R}^{d_\text{text}}$.

To obtain high-level translations for each scene entity $Q_i$, we employ a multi-head attention layer, where the input key is $e'$ and the query as well as the value are extracted point features $[Q_0, Q_1, \ldots, Q_M]$. We revise the calculation of this layer as follows:

$$\mathrm{Attention}(\mathrm{query}, \mathrm{key}, \mathrm{value}) = \mathrm{Softmax}(\frac{\mathrm{query} \cdot \mathrm{key}^\top}{\sqrt{n}})\mathrm{value},$$

$$\mathrm{MultiheadAttention}(\mathrm{query}, \mathrm{key}, \mathrm{value}) = [\mathrm{head}_1; \ldots; \mathrm{head}_h]\mathbf{W}^O \tag{28}$$

$$\text{where } \mathrm{head}_i = \mathrm{Attention}(\mathrm{query} \cdot \mathbf{W}_i^Q, \mathrm{key} \cdot \mathbf{W}_i^K, \mathrm{value} \cdot \mathbf{W}_i^V) \,.$$

Upon passing through this layer, we obtain latent features denoted as $\mathbf{z}_i \in \mathbb{R}^{d_\mathbf{v}}$ and attention weight represented as $\mathbf{w}_i \in [0, 1]$. The high-level translation of each scene entity is given by: $[\mathbf{v}_0, \mathbf{v}_1, \ldots, \mathbf{v}_M] = \mathrm{LinearLayers}([e' \parallel \mathbf{z}_0, e' \parallel \mathbf{z}_1, \ldots, e' \parallel \mathbf{z}_M]) \in \mathbb{R}^3$, in which $\parallel$ denotes as the concatenate notation. The follow-up step in LSDM involves determining the transformation matrix for each point in the scene entities by reusing the multi-head attention: $[\mathbf{F}'_0, \ldots, \mathbf{F}'_M] = \mathrm{MultiheadAttention}\left([\mathbf{v}_0, \ldots, \mathbf{v}_M], [Q_0, \ldots, Q_M], [Q_0, \ldots, Q_M]\right)$, where $\mathbf{F}'_i \in \mathbb{R}^{N \times d_\mathbf{F}}$. The output transformation matrices are computed given by: $[\mathbf{F}_0, \ldots, \mathbf{F}_M] = \mathrm{LinearLayers}\left([\mathbf{F}'_0, \ldots, \mathbf{F}'_M]\right) \in \mathbb{R}^{(M+1) \times 12}$.

Guiding points inferred from each point cloud, is calculated as $\overline{\mathbf{S}}_i = \{\mathbf{F}_{i;j} Q_{i;j}^\top | j = \overline{0; N}\} \in \mathbb{R}^{N \times 3}, \forall i = \overline{0; M}$. For simplicity, we represent the application of the transformation matrix to point $Q_{i;j}$ as $\mathbf{F}_{i;j} Q_{i;j}^\top$, followed by the equation presented in Eq. (10). The weighted guiding points are then obtained via a reduction using $\mathbf{w}$ as follows: $\tilde{\mathbf{S}} = \sum_{i=0}^M \overline{\mathbf{S}}_i \mathbf{w}_i \in \mathbb{R}^{N \times 3}$.

In order to align the hidden features of the current timestep with the size of the point cloud, we replicate them $N$ times: $t' = \mathrm{Repeat}(\mathrm{LinearLayers}(t + 1)) \in \mathbb{R}^{N \times d_t}$. Finally, the denoised point cloud is calculated by the following process:

$$\mathbf{x}'_{t+1} = \mathrm{LinearLayers}(\mathbf{x}_{t+1} \parallel t') \in \mathbb{R}^{N \times 3},$$

$$\mathbf{x}'_t = \mathbf{x}'_{t+1} + \tilde{\mathbf{S}} \in \mathbb{R}^{N \times 3}, \tag{29}$$

$$\mathbf{x}_t = \mathrm{LinearLayers}(\mathbf{x}'_t) \in \mathbb{R}^{N \times 3} \,.$$

During training, we use the loss function similar to Tevet et al. (2022):

$$\mathcal{L} = \|\mathbf{x}_0 - \hat{\mathbf{x}}_0\|^2 . \tag{30}$$

**Architecture Summarization.** As illustrated in the main paper, our network architecture contains: (*i*) a human pose backbone, (*ii*) a point cloud backbone, (*iii*) a text encoder followed by standardized MLP layers, (*iv*) a multi-head attention followed by MLP layers to compute $\mathbf{v}$, (*v*) a multi-head attention followed by MLP layers to compute $\mathbf{F}$, (*vi*) MLP layers for transformation operations, and (*vii*) an MLP encoder-decoder pair. We summarize the neural architecture as well as hyperparameters of LSDM as in Table 6 and 7.

| Component | Description | Input size | Output size |
|---|---|---|---|
| (*i*) | A pcd. backbone extracting features from the human pose | $[N, 3]$ | $[N, 3]$ |
| (*ii*) | A pcd. backbone extracting features from $M$ objects | $[M, N, 3]$ | $[M, N, 3]$ |
| (*iii*-a) | A text encoder (CLIP or BERT) | Any | $[D]$ |
| (*iii*-b) | MLP layers | $[D]$ | $[d_{\text{text}}]$ |
| (*iv*-a) | Multi-head attention to calculate translation vectors $\mathbf{v}$ | $[d_{\text{text}}], [M{+}1, N, 3]$ | $[M{+}1, d_{\mathbf{v}}]$ |
| (*iv*-b) | MLP layers | $[M{+}1, d_{\mathbf{v}}]$ | $[M{+}1, 3]$ |
| (*v*-a) | Multi-head attention to calculate transformation matrix $\mathbf{F}$ | $[M{+}1, 3], [M{+}1, N, 3]$ | $[M{+}1, N, d_{\mathbf{F}}]$ |
| (*v*-b) | MLP layers | $[M{+}1, N, d_{\mathbf{F}}]$ | $[M{+}1, N, 12]$ |
| (*vi*) | Transformation operations | $[M{+}1], [M{+}1, N, 12], [M{+}1, N, 3]$ | $[M{+}1, N, 3]$ |
| (*vii*) | MLP encoder-decoder pair | $[d_{\text{time}}], [M{+}1, N, 3], [M{+}1, N, 3]$ | $[M+1, N, 3]$ |

Table 6: **Architecture specifications.**

| Hyperparameter | Value |
|---|---|
| $N$ | 1024 |
| $M$ | 8 |
| $D_{\text{CLIP}}$ of (iii) | 512 |
| $D_{\text{BERT}}$ of (iii) | 768 |
| $d_{\text{text}}$ of (iii) | 128 |
| $d_{\mathbf{v}}$ of (iv) | 32 |
| $d_{\mathbf{F}}$ of (v) | 128 |
| $d_{\text{time}}$ of (vii) | 32 |
| Num. attention layers | 12 |
| Num. attention heads | 8 |

Table 7: **Hyperparameter details.**

# D   Dataset Construction

**PRO-teXt.** While PROXD (Hassan et al., 2019) lacks semantic segmentation of the scenes, PROXE (Zhang et al., 2020) resolves this limitation by adding semantic point clouds for room objects. In our study, we utilize the semantic scenes provided by PROXE and integrate text prompts into each human motion from the PROXD dataset. Since a human sequence mainly contains movements and only interacts with the scene at a handful of moments; therefore, for each motion, we extract 3-5 human poses and generate a text prompt that describes the interaction between the current pose and the scene. Because we add text to PROXD, we reformulate the name of this dataset as *PRO-teXt*. In total, we adapt the alignments of human motions across 12 scenes, spanning 43 sequences, resulting in a total of 200 interactions.

**HUMANISE.** The scene meshes used in HUMANISE (Wang et al., 2022) are derived from ScanNet V2 (Dai et al., 2017). As ScanNet V2 provides excellent instance segmentation, and HUMANISE aligns human motions with scenes sufficiently, we directly leverage the text prompts provided by the authors. Although these prompts are action-driven, they still contain spatial relations between target objects and scene entities and are thus suitable for our problem. As HUMANISE is a large-scale dataset, we limit our study to 160 interactions, as our paper does not primarily focus on human actions, which is the original purpose of HUMANISE (Wang et al., 2022).

In summary, we present the object distribution of both datasets in Fig. 9. Our statistics reveal that there is a diverse range of room objects in both datasets, with chairs being the most prevalent asset

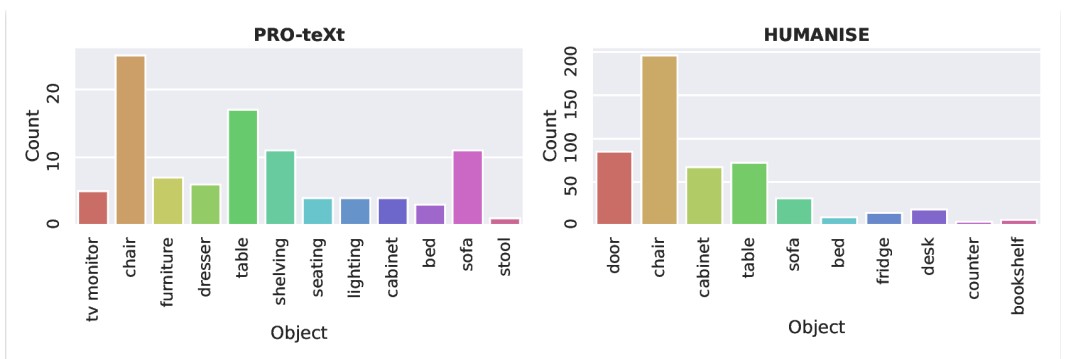

Figure 9: **Object distribution of two datasets.**

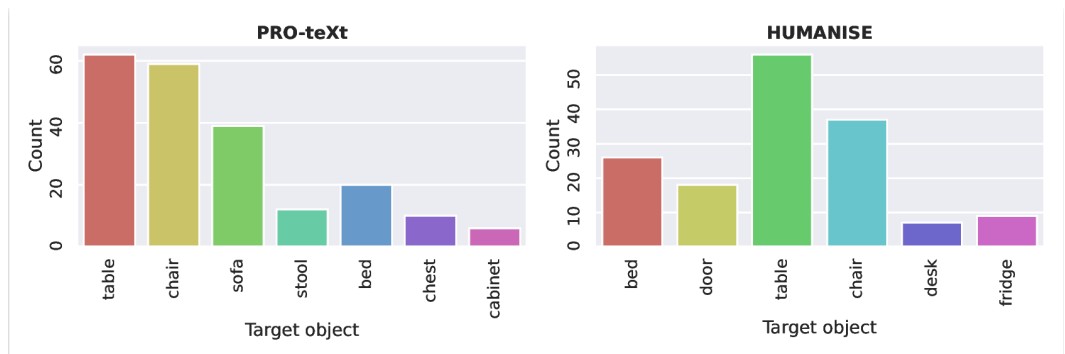

Figure 10: **Objects counted by appearances in textual commands.**

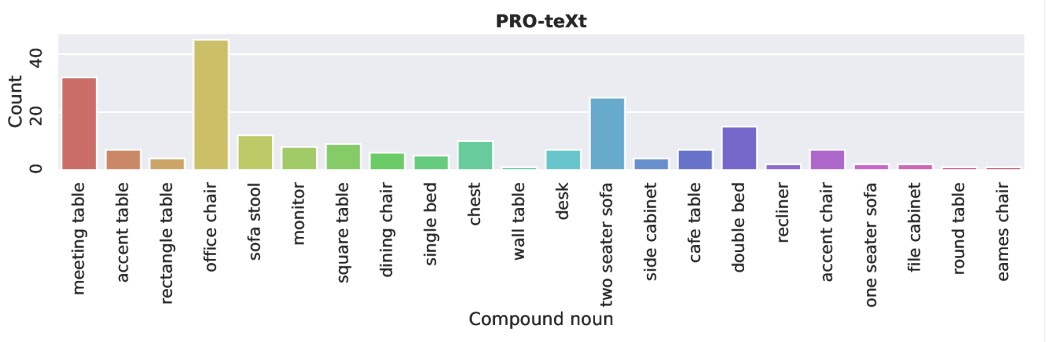

Figure 11: **Compound nouns distribution of PRO-teXt.**

in both cases. Fig. 10 gives information about the distribution of target objects. A major difference between our dataset and HUMANISE is that we describe objects as compound nouns. As a result, our text prompts comprise information about the shape of objects, which also enables scene editing operations that are not feasible with HUMANISE. We illustrate the distribution of compound nouns in Fig. 11, which demonstrates that PRO-teXt has a broad spectrum of object shapes.

# E    Scene Editing Formulation

In this section, we further elaborate on the procedure of each editing operation. We utilize a pre-trained model that was previously trained for the scene synthesis task and evaluate its performance on unseen editing interactions to assess its generalizability.

| Editing operation | Fitness ↑ | Inlier MSE ↓ | Correspondent percentage (%) ↑ |
|---|---|---|---|
| Object replacement | 0.6815 | 0.0411 | 76.37 |
| Shape alternation | 0.6130 | 0.0340 | 60.05 |

Table 8: **Ground truth evaluation.** We assess the process of constructing ground truth in editing operations.

**Object Replacement.** We first select 10 interactions out of the test set for the object replacement as well as other operations. For every interaction, we adjust the text prompt $e$ to include information about the new object category. The resulting modified prompt $e^*$ captures this information.

In the next step, we proceed to compute the ground truth object through the editing operation by selecting an object $O^*_{M+1}$ that matches the description of the new text prompt $e^*$. The alignment of $O^*_{M+1}$ with the original object $O_{M+1}$ in terms of position and rotation is performed using the ICP algorithm, subject to the constraint of fixing the $z$-rotation of the objects, as presented in (Rusinkiewicz and Levoy, 2001).

We evaluate the quality of transformation in Table 8. The metrics include: *i) Fitness score:* measures the overlapping area between the inlier correspondences and the original point cloud $O_{M+1}$; *ii) Inlier MSE:* calculates the MSE of all inlier correspondences; and *iii) Correspondent percentage:* is the ratio of the correspondence set and $O_{M+1}$. Our ground truth construction for the object replacement operation achieves significantly high scores in all metrics, in comparison to their maximum values. It is crucial to note that due to the fundamental dissimilarities in category and shape between $O^*_{M+1}$ and $O_{M+1}$, the metrics are infeasible to reach their maximum values. Based on the ground truth assessment, the position of $O^*_{M+1}$ following transformations almost matches that of the original object's position, hence can serve as a reliable ground truth for our operation.

Finally, we proceed to evaluate the object replacement operation similar to the former scene synthesis task, utilizing the established ground truths. The process involves denoising a noisy datapoint at timestep $T$, conditioning on the human pose, given objects, and the new modal command $e^*$, until we ultimately obtain the generated object $\mathbf{x}_0$. The results are presented in the main paper.

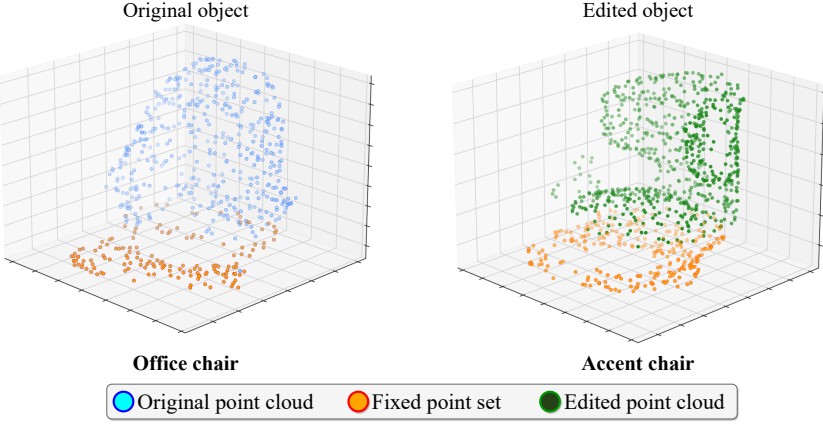

Figure 12: **Shape alternation formulation.** We fix 25% of the point that is closest to the ground (orange) then diffuse the rest 75% (blue) to obtain a new shape (green). We keep the object and spatial description in the text but change the adjective of the object (office → accent).

**Shape Alternation.** Fig. 12 demonstrates how the shape alternation operation progresses. Like the previous operation, we first construct the data and ground truth. Since this operation aims to alter the original object's shape, we replace the adjective in the original text prompt $e$ with a new shape adjective, as illustrated in the figure. The ground truth construction occurs in the same procedure as the object replacement. Table 8 exhibits the quality of the ground truth construction. With acceptable metrics, we adopt the transformed objects as supervisors for this operation.

The main difference between shape alternation and object replacement is that instead of diffusing the entire point cloud, only three-quarters of it are diffused. To ensure consistency in terms of the position and rotation of the edited object, we fix 25% of the point cloud with the lowest $z$-coordinate, *i.e.*, points that are closest to the floor (see Fig. 12). This fixation strategy is inspired by the editing application in (Tevet et al., 2022). We note that not all objects will support the shape alternation operation, however, this operation allows us to have more practical applications in real-world scenarios.

**Object Displacement.** The final operation introduced in this paper is object displacement. The formulation of this task involves modifying the spatial relation expressed in the textual input $e$, while keeping the rest of the information unchanged. For the ground truth of this operation, we manually set the correct objects semantically related to the editing text prompts.

# F  Extra Experiments

**Baselines.** We provide more information about the implementation of other baselines:

*1) ATISS* (Paschalidou et al., 2021). As stated in (Yi et al., 2023), ATISS does not take humans into account, therefore, we extend humans as an exceptional object. Furthermore, ATISS represents objects as bounding boxes defined by quadruple: position, size, angle, and category; consequently, we convert both the input and the output to bounding box representation by utilizing an algorithm from Open3D (Zhou et al., 2018). We even upscale the target object to its bounding box as ground truth so that redundant points within boxes predicted by ATISS are still counted. We utilize the same implementation of ATISS as in the original paper.

*2) SUMMON* (Ye et al., 2022). We use the pre-trained model provided by the authors for predicting contact points and freeze the model during training sessions. Next, we create a bounding box from the forecast contact points as a supplementary object and proceed similarly to ATISS.

*3) MIME* (Yi et al., 2023). We re-implement the neural architecture as presented in (Yi et al., 2023). MIME considers human motions as a bounding box with contact labels and iteratively generates the next object base on the quadruple representations of existing scene meshes (including human poses). Note that, by the time of our paper submission, both the code and dataset of MIME have not yet been made publicly available.

**Backbone Variation.** We measure the impact of different point cloud and human pose backbones in Table 9. For the point cloud backbones, we utilize PointNet++ (Qi et al., 2017b) and DGCNN (Wang et al., 2019b), two off-the-self approaches. We leverage state-of-the-art POSA (Hassan et al., 2021) and P2R-Net (Nie et al., 2022) to encode human pose. In case of representing human pose as SMPL parameters, we resolve this problem by employing a SMPL-X (Pavlakos et al., 2019) model to extract vertices from human body. Our method uses CLIP embedding (Radford et al., 2021) and BERT (Devlin et al., 2018) to encode text prompts.

| Text encoder | Human backbone | Point cloud backbone | CD ↓ | EMD ↓ | F1 ↑ |
|---|---|---|---|---|---|
| CLIP | POSA | PointNet++ | 0.5365 | **0.5906** | **0.3686** |
| CLIP | POSA | DGCNN | **0.3499** | 0.6811 | 0.3375 |
| CLIP | P2R | PointNet++ | 0.6255 | 0.9392 | 0.1288 |
| CLIP | P2R | DGCNN | 0.5131 | 0.6793 | 0.3496 |
| BERT | POSA | PointNet++ | 0.7050 | 0.9971 | 0.1148 |
| BERT | POSA | DGCNN | 0.9276 | 0.9621 | 0.2431 |
| BERT | P2R | PointNet++ | 1.7615 | 1.2433 | 0.0320 |
| BERT | P2R | DGCNN | 0.9136 | 1.0472 | 0.0534 |

Table 9: **Backbone variation.** Experiments were conducted on PRO-teXt.

Our findings indicate that CLIP is more effective when combined with both human backbone and PCD backbone in comparison to BERT. Each configuration of human backbone and PCD backbone, when integrated with CLIP, yields more comprehensive performance compared to the integration

with BERT. For instance, the combination of CLIP, P2R, and DGCNN achieves an F1 score nearly seven times higher than the combination of BERT, P2R, and DGCNN.

Moreover, it can be observed that POSA exhibits better evaluation metrics compared to P2R when paired with other components. This distinction is particularly observable when utilizing CLIP as the text encoder and PointNet++ as the PCD backbone. In this scenario, the F1 score achieved by POSA is nearly three times higher than that of P2R.

The effectiveness of PointNet++ and DGCNN is found to be comparable, with no noticeable differences observed upon replacing PointNet++ with DGCNN. Among all the possible combinations of components, the most comprehensive configuration involving CLIP, POSA, and PointNet++ achieves the the best EMD and F1 metrics, while also outputting a decent CD metric (only 0.1866 units worse than the best CD metric observed). Follow by this fact, we also select this combination as the default setting for LSDM.

| | PRO-teXt | | HUMANISE | |
|---|---|---|---|---|
| Baseline | 3D IP ↓ | FID ↓ | 3D IP ↓ | FID ↓ |
| ATISS (Paschalidou et al., 2021) | 0.0652 | 319.24 | 0.0672 | 196.83 |
| SUMMON (Ye et al., 2022) | 0.0559 | 163.98 | 0.0719 | 127.12 |
| MIME (Yi et al., 2023) | 0.0620 | 257.82 | **0.0626** | 373.02 |
| LSDM w.o text (Ours) | **0.0161** | 167.97 | 0.0768 | 94.16 |
| LSDM (Ours) | 0.0402 | **161.05** | 0.0851 | **83.96** |

Table 10: **Supplementary results on the scene synthesis task.**

**Supplementary Results.** We provide results on auxiliary metrics from all baselines in Table 10. All baselines yield statistically insignificant values in terms of 3D IP, indicating that the generated objects generated by all methods exhibit limited collision with other scene entities. However, on the FID metrics, our method remarkably improves other baselines. This improvement is particularly considerable in the HUMANISE dataset, where our method outperforms other state-of-the-art benchmarks by at least 35%.

**Impact of Number of Frames.** We remark that our model can take either single-frame human pose or multi-frame human pose. We have also included a study regarding the impact of the number of frames on scene synthesis results in Fig. 13a. The experimental results indicate that the performance of our LSDM varies little when taking different number of human pose frames as the input. Consequently, using one frame is enough for our network.

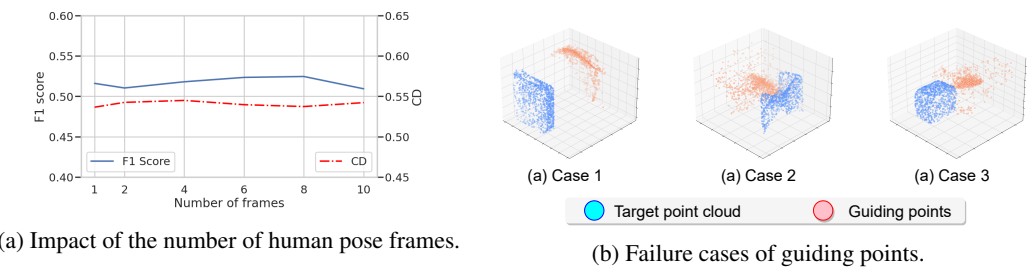

(a) Impact of the number of human pose frames.

(b) Failure cases of guiding points.

Figure 13: **Additional experiments.**

**Failure Cases of Guiding Point Network.** We visualize failure cases of predicting $\tilde{\mathbf{S}}$ of our LSDM method in Fig. 13b. Although our guiding points fail to predict the object's position, they still span over the correct object's shape.

# G  User Study Details

This section presents the implementation of our user study in detail. We conduct a user study with the participation of 40 people with a variety of ages, and professions. The male/female ratio is 50/50. For each questionnaire, we provide 15 questions about preferences, corresponding to five baselines (ground truth, LSDM, ATISS, SUMMON, and MIME). For each baseline, we anonymize the name and ask participants to evaluate three categories:

*1) Naturalness.* Naturalness of the scene is the level of familiarity that the interviewers experience when viewing the pictures in comparison to actual real-life room arrangements.

*2) Non-collision.* This metric is assessed by measuring the degree to which participants perceive objects overlapping within the scene. A lower score is assigned to the scene when participants report a greater sense of overlap among objects within it.

*3) Intentional Matching.* The final criterion we use in this study assesses the extent to which the arrangement of objects within the scene aligns with the personalized intention outlined in the text prompt.

Overall, we visualize 8 scenarios from both PRO-teXt and HUMANISE datasets for each baseline. All images and videos were collated into a single question, and interviewees were asked to score the performance of each baseline based on the visual cues presented in the 8 demonstrations. The use of 8 distinct visualizations for each baseline ensured that the study is conducted with minimal bias. Each question is rated on a scale of 1 to 5. The results are presented in the main paper.

