# OpenReview forum: "Language-driven Scene Synthesis using Multi-conditional Diffusion Model"
_NeurIPS.cc/2023/Conference — NeurIPS 2023 poster_

### Official Review · Reviewer_YsUe · 2023-06-27

**Soundness:** 3 good
**Presentation:** 3 good
**Contribution:** 3 good
**Rating:** 6
**Confidence:** 4

**Summary:**

This paper approaches the task of predicting a location and orientation of furniture, conditioned upon a person’s motion sequence, existing furniture, and text. By conditioning on text, which prior work does not do, the proposed method enables users to actively specify furniture location. In addition, experiments show it enables object replacement, shape alteration and displacement. This work introduces captions to prior human motion-furniture dataset PROXD, and shows SOTA performance given captions.

**Strengths:**

This paper adds a useful contribution to the task of human-guided scene layout
- Prior work generates furniture from motion. This work enables users to specify location using text, making it much more applicable to real-world scenarios
- Gathering captions to a standard dataset PROXD enables the proposed method to significantly outperform prior work across metrics. The paper contains good analysis of text prompts in Supplemental.
- The method also can leverage text from HUMANISE, again enabling significant improvement
- The approach also enables object editing, which is experimentally evaluated

The method proposes the intuitive approach of “guiding points” to this conditional 3D diffusion task, which it shows is highly effective in experiments.
- The idea of conditioning on a weighted combination of predicted locations from each conditioning component is intuitively more powerful than conditioning on latent encodings
- The paper provides theoretical guarantees guiding points explicitly contribute to denoising
- Experiments show conditioning upon guiding points is more effective than translation vectors alone, or unconditional
- Experiments also show correlation between accuracy of guiding points and final performance, empirically confirming theoretical findings.


**Weaknesses:**

Edit: after rebuttal, my concerns are well-addressed.

There are several missing details and comparisons that make full assessment of the paper challenging. This includes missing limitations.
- I do not understand how the training works for scene synthesis, which makes it hard to fully assess the importance of guiding points. My understanding is the LSDM denoises a point cloud given the output of the guiding points network. I assume then, the guiding points network is trained jointly, end to end with the LSDM, at each denoising step? And that no networks are pretrained, including text encoder (I’d assume this is pretrained)? This would mean S is not actually trained to predict final position, but rather consists more of geometric features? Based on Figure 6, it is hard to determine if S is directly supervised.
- It feels like another reasonable design choice would be to initialize diffusion with guiding points, and denoise these, as opposed to (or in addition to) conditioning upon them. Testing this choice could perhaps more directly validate the theoretical findings that using S specifically for conditioning is helpful.
- In qualitative results, a single human location is used. However, the proposed task is to consider a vector of human locations (“motion”). How does text work given the input is not a single location, but a set of locations? Text is sensitive to location e.g. “Place a desk in front of me”. Is time assumed to be the last timestep? In this case, does the dataset generate text descriptions based only on the last location of the human? In reality, I would imagine users would like to specify text based on any number of timesteps throughout the trajectory e.g. “place the desk in front of me [at frame i<N of N]”
- The comparison to multi-conditional modeling is not fully satisfying. The method compares to itself without F, but otherwise keeps the same geometric-based architecture. Namely, it combines weights linearly using w. A more standard conditional diffusion approach would be to concatenate features or combine them through nonlinear (e.g. transformer) layers. This comparison would make the argument for the proposed method stronger.
- Is there a breakdown into in-contact vs. not in-contact objects? Prior work specifically uses this; as this paper claims to outperform in not-in-contact objects, it would be a helpful metric to report.

Contributions feel slightly niche (minor weakness)
- The central contribution of conditioning upon a geometrically transformed linear combination of feature distances of objects and humans is a cool contribution. However, it feels specific to the task of furniture placement conditional on human motion and text. Is there a wider reason this method is important?
- Saying scene synthesis has gained significant attention in the past few years and citing one paper from last year with 3 citations (L16) is not convincing the task is very important
- Text-conditioned diffusion models predicting position and orientation already exist in the near subfield of human motion generation (Tevet et al. Human Motion Diffusion Model, ICLR 2023). The application in the near subfield of furniture position and orientation on its own feels like a relatively modest step.


**Questions:**

-	L4: “combing” -> combining
-	L51: “a new challenging” -> “a new challenge”


**Limitations:**

No. I would recommend particular focus on the ability to utilize a sequence of human motion (see weaknesses), assuming this is a limitation. Others could include the assumption one knows the object of interest and further has a 3D model of it.

---

> ### Author Rebuttal · Authors · 2023-08-04
>
> Thank you for your valuable review.
>
> **Q1: How the training works for scene synthesis? The guiding points network is trained jointly, end to end with LSDM?**
> >The key reason why the training works is because of our theoretical findings. Eq. (7) shows $\tilde{S}$ explicitly contributes to the denoising process, supporting the assumption that $\tilde{S}$ would also be learned concurrently with the denoising process. We trained the guiding point network end-to-end with LSDM. Only the text encoder is pretrained.
>
> **Q2: S is not actually trained to predict final position but rather consists more of geometric features? It is hard to determine if S is directly supervised.**
> >$\tilde{S}$ is not just simply geometry feature; it was designed to forecast the target object (please see our Appendix's Motivation). In fact, because $\tilde{S}$ is not directly supervised, we have to carefully examine whether this quantity is meaningful (L249). The theory and experiments (Corollary 1.2, Table 4, Figure 6 in the main paper, and additional failure cases in Figure 3 of the One-page PDF) further show that $\tilde{S}$ represents a good approximation for the target object.
>
> **Q3: Why not initializing diffusion with guiding points, and denoise these?**
> >There are three reasons why initializing diffusion with guiding points is not a reasonable design choice. First and foremost, the initial state for the denoising process is sampled from an approximation of isotropic Gaussian distribution [1, 2, 3] rather than a specific quantity; therefore, initializing the denoising process with the guiding points seems not popular. Consequently, conditioning upon guiding points seems to be a more natural approach. Second, as we have indicated in Eq. (7) that denoising will also concurrently benefit the learning of the guiding points; therefore, our end-to-end approach has a rationale behind its solid performance. Finally, training in two stages requires longer time and computational resources.
>
> **Q4: Experiment regarding i) initialize diffusion with guiding points, and denoise these and ii) comparison to multi-conditional modeling (concatenate features or combine them through nonlinear (e.g. transformer) layers.**
> >Following your suggestion, we have implemented the two-stage version: one stage is for supervising the guiding points, then the second stage is to utilize the pretrained guiding points network; and another multi-conditional diffusion model (MCDM) that concatenates all of the latent features extracted from the conditions (text prompt, scene entities) and pass through a transformer layer. The results show that LSDM clearly outperforms both the two-stage method and MCDM. Note that, training two-stage LSDM took about 1.5 times longer than ours.
>
> |||PRO-teXt|||HUMANISE||
> |-|-|-|-|-|-|-|
> | Baseline|CD|EMD|F1|CD|EMD|F1|
> | MCDM|0.630|0.726|0.357|0.858|0.875|0.251|
> | Two-stage LSDM|0.562|0.621|0.437|0.744|0.806|0.353|
> | LSDM (ours)|**0.536**|**0.590**|**0.516**|**0.737**|**0.750**|**0.439**|
>
> **Q5: How does text work given the input is not a single location, but a set of locations?**
> >If the input includes a set of locations, our method will consider each location as a condition and generate new objects based on the current human location, given objects, and current text prompt condition. However, this case has not been intensively tested in our experiment due to the lack of training data following this scenario.
>
> **Q6: Is time assumed to be the last timestep? Does the dataset generate text descriptions based only on the last location of the human?**
> >We do not assume that text descriptions are based only on the last human location when creating dataset. Indeed, we label the dataset as your observation (L98-99 of the Appendix), and we allow the user to "specify text based on any number of timesteps throughout the trajectory e.g. place the desk in front of me [at frame i<N of N]".
>
> **Q7: Is there a breakdown into in-contact vs. not in-contact objects?**
> >We break down the contact and non-contact results in PRO-teXt in the table below. Our LSDM achieves better performances in both cases.
>
> |||Contact|||Non-contact||
> |-|-|-|-|-|-|-|
> |Baseline|CD|EMD|F1|CD|EMD|F1|
> |ATISS|0.779|1.018|0.128|3.248|1.619|0.026|
> |SUMMON|0.780|1.001|0.139|3.324|1.600|0.028|
> |MIME|0.717|0.978|0.145|3.179|1.597|0.024|
> |LSDM (ours)|**0.081**|**0.433**|**0.703**|**0.915**| **0.737**|**0.471**|
>
> **Q8: Is there a wider reason this method is important?**
> >In terms of theory, we hope that our proposed guiding point is a general concept and may be useful for other tasks. For example, in the visual grounding task, we can predict "guiding pixels" indicating possible boxes from the conditions to guide the denoising process. In terms of application, our method can be used in animation, metaverse, gaming.
>
> **Q9: About the importance of scene synthesis task and writing in the Introduction.**
> >We appreciate your comments and have revised the Introduction to stress more the importance of the scene synthesis task.
>
> **Q10: Comparison with Text-conditioned diffusion models (Tevet et al).**
> >Tevet et al. use text as the *sole input* to generate *human motion* while we utilize *multi-condition* (text, human, objects) to generate *new objects*. We believe there is a significant difference in theory and application between two works.
>
> **Q11: About the ability to utilize a sequence of human motion?**
> >Our method can take input as a human motion sequence. Figure 4 in the One-page PDF shows that the sequence of human motion indeed does not bring significant improvement.
>
> >All typos have been fixed. Thanks!
>
> References:
>
> [1] Ho et al. Denoising diffusion probabilistic models. NeurIPS 2020.
>
> [2] Dhariwa et al. Diffusion models beat gans on image synthesis. NeurIPS 2021.
>
> [3] Tevel et al. Human motion diffusion model. ICLR 2023.

---

> > ### Author Response · Authors · 2023-08-15
> > **Please let us know if you have any further concerns**
> >
> > Dear Reviewer YsUe,
> >
> > Thanks for your constructive efforts in the reviewing process of our paper! Let us know if you have any further concerns before the end of the discussion phase.
> >
> > Thanks, Authors.

---

> > ### Comment · Reviewer_YsUe · 2023-08-15
> > **Reviewer Response to Rebuttal**
> >
> > After reading the rebuttal and other reviews I change my rating to 6 - weak accept.
> >
> > I believe the paper should be accepted as (1) I agree with the other authors adding text conditioning to human-guided scene layout is an important contribution; (2) contributions of the method, such as the interesting "guiding points" are well-defended in experiments; and finally (3) clarifying my concerns about training details, denosing design choices, etc.

---

> > > ### Author Response · Authors · 2023-08-22
> > > **Thanks for your reconsideration**
> > >
> > > Dear Reviewer **YsUe**,
> > >
> > > We would like to express our appreciation for your thoughtful reconsideration! Best regards, Authors.

---

### Official Review · Reviewer_c9eB · 2023-07-04

**Soundness:** 3 good
**Presentation:** 3 good
**Contribution:** 3 good
**Rating:** 7
**Confidence:** 4

**Summary:**

This paper focuses on language-driven scene synthesis, a new task integrating text prompts, human motions, and existing objects as multiple conditions. The proposed task is challenging as it requires a strategy for encoding the multi-modal conditions into a unified space. To solve the problem, the authors introduce a novel guiding points concept to combine multiple conditions, which can explicitly contribute to the denoising process. They also introduce three scene-editing applications based on the text prompt input. They demonstrate the approach empirically and theoretically; the intensive experiments show that the proposed approach achieves significant improvements over the state-of-the-art methods.

**Strengths:**

1. Extending the scene synthesis to a language-driven setting, incorporating text prompts and human motions as input, holds great promise and significance in bridging the gap between research and real-world applications. It also enables downstream real-world scene editing applications.

2. This paper proposes a somewhat novel method to handle such a multi-conditional setting. The authors introduce guiding points that explicitly guide the reverse process of the diffusion model, offering a departure from the implicit unification approach used in previous multi-conditional diffusion models.

3. This paper's theoretical demonstration and experimental analysis are comprehensive, especially the ablative experiment, which demonstrates the impact of different modalities and how the proposed modules contribute to the overall performance.


**Weaknesses:**

1. In comparing MIME to your approach, I notice that MIME focuses on generating 3D scenes based on 3D human motion, whereas your method takes a human pose as input. Considering this distinction, is it fair to make a direct comparison between MIME and your approach?

2. It has come to my attention that Proposition 2 and Corollary 2.1 are included in section 3.2. However, it may be more suitable to relocate them to the supplementary materials. I am uncertain about the significance of these components within the section, and it seems that the author included them primarily to showcase their expertise.

**Questions:**

1. Could you please elaborate on the methodology used to convert the point cloud into an object mesh shown in your figures and video?

2. I'm curious about the performance of directly learning the unification of multiple conditions with a diffusion model. It appears that the proposed method outperforms others, but what about the performance gap between these two methods?

3. Based on my comprehension, does your model exclusively utilize a single-frame human pose as input rather than a motion sequence? If that is the case, I'm curious about the rationale behind this choice, considering that a motion sequence could potentially provide more comprehensive information about the scene distribution. By the way, how do you represent the human body?


Minor fixes:

- The video in the supplementary material is excellent, but you can provide some visualization of raw point cloud results.
- Authors are advised to provide a limitation and future work section.

**Limitations:**

The authors have not discussed this paper's limitations and societal impact.

---

> ### Author Rebuttal · Authors · 2023-08-04
>
> Thank you for your valuable feedback and insightful review.
>
> **Q1: As MIME utilizes a sequence of human motions, your method only utilizes a single-frame motion. Does your model exclusively utilize a single-frame human pose as input rather than a motion sequence?**
> >Yes. We believe this distinction is minor in our problem settings because the target objects are conditional constrained only on the *moment* the user commands, and they little depend on past motions of the human. Therefore, the final human pose before executing the command gives adequate information to complete the task. Thus, no unfair comparison was taking place though the input may be different.
>
> **Q2: Is it fair to make a direct comparison between MIME and your approach?**
> >We remark that our model can take either single-frame human pose or multi-frame human pose. We have also included a study regarding the impact of the number of frames on scene synthesis results in Figure 4 of the One-page PDF. The experimental results indicate that the performance of our LSDM varies little when taking different number of human pose frames as the input. Consequently, using one frame is enough for our network. In addition, we also implement a variation of MIME with text prompt by concatenating CLIP text encoder features with the latent features at the transformer layer of MIME's architecture. The result is as follows.
>
> |||PRO-teXt|||HUMANISE||
> |-|-|-|-|-|-|-|
> |Baseline|CD|	EMD|F1|CD|EMD|F1|
> |MIME|2.0493	|1.3832|0.0990|5.4259|2.0837|0.0628|
> |MIME with text|1.8424|1.2865|0.1032|4.7035|1.8201|0.0849|
> |LSDM (ours)|**0.5365**	|**0.5906**|**0.5160**|**0.7379**|**0.7505**|**0.4395**|
>
> **Q3: What is the rationale behind this choice?**
> >There are two key reasons why we only take a single-frame human pose as input. First, the dependency between text prompts and human motions is temporal, i.e., the semantics of placing objects depends on one and only one frame. In fact, if there is a text prompt "Place a table in front of me" and multiple frames, it would be ambiguous to determine which moment the table refers to. Second, the human pose frame referenced to the prompt gives adequate information, which is indicated in Section 3 of our Appendix and Figure 4 in our One-page PDF.
>
> **Q4: It has come to my attention that Proposition 2 and Corollary 2.1 are included in section 3.2. However, it may be more suitable to relocate them to the supplementary materials.**
> >Thank you for your suggestion. As suggested by you and Reviewer **uwsL**, we have shortened Remark 1.2 and moved Proposition 2 + Corollary 2.1 to the Appendix.
>
> **Q5: Significance of Proposition 2 and Corollary 2.1?**
> >Corollary 2.1 provides a reliable measurement for evaluating guiding points. L152 indicates an important observation; that is, a smaller MSE between the predicted guiding points $\tilde{S}$ and $\mu_0$ corresponds to a more accurate estimation. This observation gives sufficient evidence to conclude from Table 4 that our guiding points are meaningful.
>
> **Q6: Could you please elaborate on the methodology used to convert the point cloud into an object mesh shown in your figures and video?**
> >We utilize the object recovery algorithm in [1] (Line 212 in our main paper). The key idea is to iterate through all possible objects and then determine the one that aligns the most with the point cloud.
>
> **Q7: Performance of directly learning the unification of multiple conditions with a diffusion model?**
> >We have implemented another multi-conditional diffusion model (MCDM) to directly unify all the conditions and pass the latent features through a transformer layer. The performance of this latent mechanism still lags behind our LSDM's. The results are reported in the following table.
>
> |             |            | PRO-teXt   |            |            | HUMANISE   |            |
> |-------------|------------|------------|------------|------------|------------|------------|
> | Baseline    | CD         | EMD        | F1         | CD         | EMD        | F1         |
> | MCDM        | 0.6308     | 0.7269     | 0.3579     | 0.8583     | 0.8757     | 0.2505     |
> | LSDM (ours) | **0.5365** | **0.5906** | **0.5160** | **0.7379** | **0.7505** | **0.4395** |
>
> **Q8: How do you represent the human body?**
> >We use point cloud or SMPL models as in several works, e.g., [1, 2, 3] (Line 179 in our Appendix). For the HUMANISE dataset, we use SMPL models to represent humans. For the PRO-teXt dataset, the input for human motion is a 3D point cloud, which we follow the practice by [1].
>
> **Q9: You can provide some visualization of raw point cloud results.**
> >We provide the visualization in Figure 1 of our attached One-page PDF. Please see the attached file.
>
> **Q10: Authors are advised to provide a limitation and future work section**
> >Thank you for your comments. The limitation of our method is that the theoretical findings have an assumption constrained to uniform data like point clouds. The predicted guiding points are not always aligned with the target object, which is indicated in the results of Tables 1 and 4 of the main paper. Furthermore, the editing results show necessary improvements in future works. The broader impact of our paper lies in the potential applications in VR, animation, and metaverse. We have included these details in our paper.
>
> References:
>
> [1] Ye et al. Scene synthesis from human motion. In SIGGRAPH Asia 2022.
>
> [2] Kocabas et al. PARE: Part attention regressor for 3D human body estimation. In ICCV 2021.
>
> [3] Rempe et al.. Humor: 3d human motion model for robust pose estimation. In ICCV 2021.

---

> > ### Comment · Reviewer_c9eB · 2023-08-15
> > **Thanks for the reply**
> >
> > Thanks for your response. I have no further questions.

---

> > > ### Author Response · Authors · 2023-08-15
> > > **Thanks**
> > >
> > > Thank you for your comments!

---

### Official Review · Reviewer_RVaZ · 2023-07-04

**Soundness:** 2 fair
**Presentation:** 2 fair
**Contribution:** 2 fair
**Rating:** 4
**Confidence:** 3

**Summary:**

This paper targets to generate 3d scene  by conditioning on text prompts and other inputs, e.g., room layouts. For this purpose, they operate on 3d point cloud representation and propose a multi-conditional diffusion model to generate guiding points to achieve 3d scene synthesis purpose. The experiments are evaluated on synthetic indoor dataset.

**Strengths:**

- Adopting human pose into 3d scene generation process is a novel condition to consider during generation.

**Weaknesses:**

- The authors did not motivate well on why a diffusion model is necessary or better for this task. Given the large amount of prior work in scene layout, is there any advantage of diffusion model, such that it can do something prior method cannot?
- Their dataset is too simple. On one hand, the authors deal with 3D point cloud representation, which usually noisy and sparse in real-world scanning data. On the other hand, they only test the solution on synthetic dataset, which seems to be in a different distribution with real-world scan. An evaluation on real-world dataset can make the world more solid.
- Their video supplementary is confusing in terms of what kind of application that are aiming at. Is the audio cut off accidentally in the mp4? As for the application, is that the authors hope to leverage human pose to generate 3d objects in the indoor scene?

**Questions:**

1. Could you justify more on the method choice of diffusion model in this task?
2. The citation format is not the one suggested by NeurIPS 2023. Please check submission website to adjust in later revision.
3. What application is the proposed technique aiming to realize?

**Limitations:**

good.

---

> ### Author Rebuttal · Authors · 2023-08-04
>
> We greatly appreciate your valuable feedback and thoughtful review. Please see below our responses and let us know if you have any further questions.
>
> **Q1: Could you justify more on the method choice of diffusion model in this task?**
> >There are three reasons to leverage the diffusion model in our paper. First, prior methods in the literature *do not consider a comprehensive set of conditions* as ours, for example, some of them do not consider text prompt while others do not consider human motion. Second, diffusion models are exceptional in terms of conditional generation, which have been observed in many successes such as [1], [2]. Third, point clouds can be viewed as particles in a thermodynamic system [3], therefore, it is natural to apply diffusion probabilistic models [4].
>
> **Q2: Is there any advantage of diffusion model, such that it can do something prior method cannot?**
> >The key advantage we take from diffusion models is their strong *guidance* ability on given conditions. We justify in our paper that the guidance of our method is theoretically supported (in Remark 1.2 of section 3.2). As the object space is sparse and agnostic, a guidance strategy (in our case, the guiding point network) would give prior information about the possible span and shape of the target object, effectively guiding the network to diffuse the rest. We further remark that in [5], the authors establish a more robust baseline with a text-guidance strategy than other state-of-the-art generative models (including GAN).
>
> **Q3: Their dataset is too simple and they only test the solution on synthetic dataset.**
> >The datasets we used currently are the most recent datasets in this field. Recent work, such as ATISS, SUMMON, and MIME, also only utilized synthetic datasets, not real-world environments. Our considered datasets are based on HUMANISE and PROXD, which are also widely studied in this field [6]. We agree with you that testing more real-world datasets would be more meaningful; however, we do need to wait for such a feasible dataset.
>
> **Q4: Their video supplementary is confusing in terms of what kind of application that are aiming at and what application is the proposed technique aiming to realize?**
> >Our problem has the potential to apply to character animation or metaverse, where embodied agents can interact and give commands to generate objects that are aligned with the scene's spatial arrangement and user preferences. We have included a Broader Impact section to discuss the applications of our paper. Particularly, our technique can be applied when a user is entering an empty apartment and giving commands to automatically generate objects (e.g., "putting a table next to the bed", "placing a sofa behind the chair") to arrange the furniture (where physical contact is not mandatory).
>
> **Q5: Is the audio cut off accidentally in the mp4?**
> >Our video does not include audio in this version.
>
> **Q6: As for the application, is that the authors hope to leverage human pose to generate 3d objects in the indoor scene?**
> >Yes, we believe that the generation of objects conditioned on user preferences (such as text prompts and human pose) can be applied to animation, metaverse, or gaming.
>
> **Q7: About the citation format.**
> >Thank you for your suggestion. The citation style has been revised in our final version.
>
> References:
>
> [1] Tseng et al. Edge: Editable dance generation from music. In CVPR 2023.
>
> [2] Tevet et al. Human motion diffusion model. In ICRL 2023.
>
> [3] Luo and Hu. Diffusion probabilistic models for 3d point cloud generation. In CVPR 2021.
>
> [4] Ho et al. Denoising diffusion probabilistic models. In NeurIPS 2021.
>
> [5] Dhariwal and Nichol. Diffusion models beat gans on image synthesis. In NeurIPS 2021.
>
> [6] Yi et al. Human-aware object placement for visual environment reconstruction. In CVPR 2022.

---

> > ### Author Response · Authors · 2023-08-15
> > **Looking forward to your response**
> >
> > Dear Reviewer RVaZ,
> >
> > Thanks for your endeavors in the reviewing process! Please let us know if you have any further questions before the end of the author-reviewer discussion phase.
> >
> > Thanks, Authors.

---

### Official Review · Reviewer_aT28 · 2023-07-06

**Soundness:** 3 good
**Presentation:** 3 good
**Contribution:** 3 good
**Rating:** 5
**Confidence:** 4

**Summary:**

This paper deals with scene synthesis with human pose, room layout, and text prompts.
The main architecture is a multi-conditional diffusion model, which performs progressive generation, where a new object is synthesized and conditioned on the existing scene point cloud and the language description.
The key contribution is a guiding point network, which first generates a reference point cloud as a weighted sum of existing objects and human pose, then the reference point cloud is used as a condition to guide the denoising processing for a new object.
The trained network allows scene generation guided by language and can produce semantically meaningful scene edits.


**Strengths:**

The progressive generation of scenes guided by language makes it much easier to interact with the synthesis process and alleviates the control burden from the designer's side.
The experiments show the effectiveness of the proposed pipeline and the learning objective.
The ablation study is interesting.

**Weaknesses:**

The architecture is quite intuitive, but the derivation is not quite clear and seems disconnected from what the author wants to do.
Equations (2) and (3) are standard, but starting from equation (4), when the guiding point is introduced, some unsureness kicks in.
For example, why do you assume that x_0 is a uniform distribution over a domain S, how do you define S in the first hand, and why x_0 be uniform is a good assumption? Also, why q(y|x_0) is non-zero uniform over S?
What is the difference between S and S_hat, and what do you mean by sampling set of x_0?
Why then q(x_0) becomes uniform?
How do you infer q(x_0|y) is also uniform over S? how?
If q(x_0) is uniform, then \mu_0 is the center of the region S? why this is a meaningful quantity in your consideration?
Why is S_tilt a sampling set of S_hat?
Eq. (10) is very intuitive, why do we need all the previous derivations?


**Questions:**

See above.

**Limitations:**

No limitation is discussed.

---

> ### Author Rebuttal · Authors · 2023-08-04
>
> Thank you for your insightful review and valuable feedback.
>
> **Q1: Why do you assume that $x_0$ is a uniform distribution over a domain $S$?; and Why $x_0$ be uniform is a good assumption?**
> >Our assumption is based on the fact that we uniformly sample the point cloud out of each object. For your follow-up question, uniform sampling is well-suited for our problem settings (3D point clouds) for three reasons. The first reason is that since we highly focus on objects' spatial arrangement, uniform-sampled point clouds are ideal for capturing the high-level geometry of the target objects [1]. Second, uniform sampling for 3D point clouds has been widely employed in previous works such as [1], [2], [3]. Third, uniform sampling is a computationally efficient strategy when compared to alternative methods; for example, uniform sampling has O(N) computational complexity, while Poisson disk sampling has O(N log N) complexity in typical cases [4].
>
> **Q2: How do you define $S$ in the first hand?**
> >$S$ represents the space of the interior of the object $O_{M+1}$. We have included this definition at the beginning of section 3.2 (L120). Thank you for bringing this to our attention.
>
> **Q3: Why $q(y|x_0)$ is non-zero uniform over $S$?**
> >$q(y|x_0)$ is non-zero uniform over $S$ under the assumption of Remark 1.2 (L131). This assumption derives from the fact that as long as we uniformly sample $x_0$ out of $S$, $x_0$ serves as a geometry representation of the target object. The meaning of the conditions $y$ (including text prompt and other scene entities) remains unchanged as they only depend on the target object's geometry. When $x_0$ is sampled from $S$, $q(y|x_0)$ is non-zero as the alignment of the conditions with the target object is meaningful.
>
> **Q4: What is the difference between $S$ and $\hat{S}$? and What do you mean by sampling set of $x_0$?**
> >$\hat{S}$ is a discretized set of the continuous space $S$. Similarly, when we refer to a "sampling set of $x_0$", we mean a discretized set of $x_0$.
>
> **Q5: Why then $q(x_0)$ becomes uniform?**
> >$q(x_0)$ is already uniform under our assumption of Remark 1.2 (please see Line 131 of our main paper).
>
> **Q6: How do you infer $q(x_0|y)$ is also uniform over $S$?**
> >Thank you for pointing this out. However, upon further investigation, we found that the uniform property of $q(x_0|y)$ is not necessary for the construction of Eq. (7) and Corollary 2.1. Therefore, we have removed this sentence.
>
> **Q7: If $q(x_0)$ is uniform, then $\mu_0$ is the center of the region $S$?**
> >The notation $\mu_0$ in L137 is originally used to denote the predicted mean of the initial probability distribution $q(x_0)$. However, the notation $\mu_0$ in L143 is used to denote the centroid of $S$. To avoid confusion and potential misunderstandings, we have made a correction by changing the notation in L137 from $\mu_0$ to $\tilde{\mu}_0$. Regarding your question, when $q(x_0)$ is uniform, the predicted mean $\tilde{\mu}_0$ indeed represents the center of the region $S$.
>
> **Q8: Why this is a meaningful quantity in your consideration?**
> >The predicted mean $\tilde{\mu}_0$ is meaningful to our paper because $\tilde{\mu}_0$ is the connection between theory and our network design. In theory, $\tilde{\mu}_0$ is an estimation of $x_0$. In the motivation for the network design (section 3 of the Appendix), we show that by applying transformation matrices to scene entities, we can predict the centroid of target object, which is formulated as $\tilde{\mu}_0$ in this case. We further remark that the estimation term $\tilde{S}$ is not restricted to the design choice of $\tilde{\mu}_0$ and can be designed differently in other tasks.
>
> **Q9: Why is $\tilde{S}$ a sampling set $\hat{S}$?**
> >$\tilde{S}$ is defined as the sampling set of predicted $\tilde{\mu}_0$ (L138); therefore, serves as an estimation for $\hat{S}$. $\tilde{S}$ is not a sampling set of $\hat{S}$.
>
> **Q10: Eq. (10) is very intuitive, why do we need all the previous derivations?**
> >Eq. (10) is not adequate for our central claim that guiding points explicitly contribute to the denoising process; therefore, we need more explicit interpretation as in Eq. (7). From Eq. (7), we observe that the term $\tilde{S}$ has an explicit contribution to the denoising process, leading to the assumption for our network architecture: guiding points $\tilde{S}$ can be learned concurrently with the denoising process. Consequently, we design our network to learn guiding points with the denoising process jointly. Experiments (Table 1, Table 4, and Figure 6) confirm this assumption. We further establish Corollary 2.1 to achieve a reliable measurement for evaluating guiding points. L152 indicates an important observation for our measurements; that is, a smaller MSE between the predicted guiding points $\tilde{S}$ and $\mu_0$ corresponds to a more accurate estimation.
>
> **Q11: No limitation is discussed.**
> >The limitation of our method is that the theoretical findings have an assumption constrained to uniform data like point clouds. The predicted guiding points are not always aligned with the target object, which is indicated in the results of Tables 1 and 4 of the main paper. Furthermore, the editing results show necessary improvements in future works. We have included a Limitation section in our final version.
>
> References:
>
> [1] Qi et al. Pointnet: Deep learning on point sets for 3d classification and segmentation. In CVPR 2017.
>
> [2] Yu et al. Pu-net: Point cloud upsampling network. In CVPR 2018.
>
> [3] Lyu et al. A conditional point diffusion-refinement paradigm for 3d point cloud completion. In ICLR 2022.
>
> [4] Yuksel, C. Sample elimination for generating Poisson disk sample sets. In Eurographics 2015.

---

> > ### Author Response · Authors · 2023-08-15
> > **Let us know if you have any further questions**
> >
> > Dear Reviewer aT28,
> >
> > Thanks for your efforts in the review! Please let us know if you have any further concerns before the end of the discussion phase.
> >
> > Thanks, Authors.

---

### Official Review · Reviewer_uwsL · 2023-07-10

**Soundness:** 3 good
**Presentation:** 1 poor
**Contribution:** 3 good
**Rating:** 6
**Confidence:** 3

**Summary:**

In this paper, the authors propose a new task named language-driven scene synthesis. This new task takes text prompts, human motion, and existing objects to generate the next object in the scene. To handle the multiple conditions, they design a guiding points strategy to unify them. It first explicitly predicts a "pseudo" target point cloud from the conditions and then uses these predicted points as a guide for the diffusion model to predict the "truly" target point cloud. They demonstrate their approach is theoretically supportive. In the experiment, they show that their method outperforms the state-of-the-art baselines. Furthermore, they introduce three scene editing tasks that are useful for application.

**Strengths:**

- The proposed language-driven scene synthesis task integrates text prompts, human motion, and existing objects as conditions. It is an interesting direction that injects user preference for scene synthesis and thus enables real-world scene editing applications with text prompts.
- To handle the multiple conditions, the authors revisit point cloud representation and propose a guiding point concept to use the conditions explicitly. They first predict a "pseudo" target point cloud from the conditions and then use these predicted points to guide the diffusion model to predict the "truly" target point cloud. This explicit strategy injects a strong inductive bias to utilize all the conditions for placing the next object.
- The experiment part is intensive and demonstrates the proposed method with text prompts, human motion, and existing objects as conditions achieve the best results compared with baselines.

**Weaknesses:**

- I have a question regarding the application of the proposed new tasks. When we take only human motion as a condition for scene synthesis, MIME (Yi et al., 2023) treat this as "turning human movement in a "scanner" of the 3D world." In your proposed task that uses human motion and text prompts, I understand it is useful when we want to place a table in the VR setting. However, what is the use case if the human motion is setting down and using "put a chair under the human" as a prompt? We can not use this in VR since we can not sit without a real chair.
- I don't like the presentation of this paper. The reasons are as below.
  1) Section 3.2 seems to break the flow of the whole paper. After reading this subsection, I need to back to Section 3.1 multiple times to remind myself of the notation for Section 3.3. It is suggested to make the theoretical support in the main paper shorter and at a high level and move the others to the supplement.
  2) Since Section 3.2 uses too much space in the main paper, the authors make Section 3.3 short and unclear. However, this is the main contribution of this paper. It is messy for the audience to read the operations with only unclear text descriptions (also without any shape information for the variables). It is suggested to add equations or pseudocode to describe the operations.
  3) Figure 2 is also unclear. For example, for the text, it is stated that "the input key is the text embedding e′, the input queries are the given scene entities." However, in Figure 2, the text embedding e′ and the scene entities are concatenated and then fed to the attention modular.
  4) I read the supplement. The motivation part deserves to be moved to the main paper. The implementation details also need to be clarified. Especially, Table 2 is unreadable. Why not list the equations of the operations?
- For the baselines, can you add text prompt conditions to MIME for your proposed task for a fair comparison? Considering their method is transformer-based, it should be easy to add text conditions.
- For the editing tasks, is the target object necessary to be the M+1 object? In the text prompt, the target object is already indicated. In this case, it seems that we can change any object in the scene instead of only the last one.
- In Line 171, you claim that "we extract spatial information from the text prompt by utilizing the off-the-shelf multi-head attention layer." What is the meaning of "off-the-shelf" here?
- In Table 2 of the supplement, the output of the text encoder is 1D. I remember that the output of the CLIP text encoder is a list of tokens. Do you apply pooling here?

**Questions:**

Please refer to the weakness section.

**Limitations:**

It seems that the authors do not properly discuss the limitation and the broader impact of their work.

---

> ### Author Rebuttal · Authors · 2023-08-04
>
> Thank you for your thoughtful review and valuable feedback.
>
> **Q1: Application of the proposed new tasks? And the "chair" user case in VR.**
> >We acknowledge and agree with your example regarding the feasibility of the chair in VR settings, as users do not physically sit on the chair. However, we believe our proposed task is still helpful in applications with non-contact objects. For example, our method can be applied when a user is entering an empty apartment and giving different commands (e.g., "putting a bed in the corner," "placing a table next to the sofa") to arrange the furniture (where physical contact is not mandatory). Using natural language input allows scene synthesis not to rely solely on human motions, which is widely assumed in previous works [1, 2]. Furthermore, our proposed tasks hold significant potential in alternative applications like animation or metaverse. In these contexts, users can control embodied agents to interact with synthesized objects, including the chair in your question (e.g., the char will be generated for the animated character in a metaverse environment and not necessarily need a physical chair for the real user).
>
> **Q2: About the writing of Section 3.2 and Section 3.3.**
> >We genuinely appreciate your insights, particularly regarding the length of section 3.2. Following your suggestions, we have shortened Remark 1.2 and moved Proposition 2 + Corollary 2.1 to the Appendix. The Motivation and Guiding Point network section from the Appendix are moved to the main paper. We have included additional equations in Section 3.3 to provide a step-by-step description of the implementation details of our method in the revised version.
>
> **Q3: About Table 2 in the Appendix.**
> >We have included a column in Table 2 to explain the functionality of each component. Below is a brief revision of this table.
>
> | Component | Description                                           	| Input shape          	| Output shape      	|
> |-----------|-----------------------------------------------------------|-------------------------|----------------------|
> | (i)   	| A human pose backbone extracting features from human motion | N x 3              	| N x 3           	|
> | (ii)  	| A point cloud backbone extracting features from M objects 	| M x N x 3           	| M x N x 3        	|
> | (iii-a)   | A text encoder (CLIP or BERT)                           	| Any                 	| 1 x D              	|
> | ... |...|... | ...|
>
> **Q4: About Figure 2 in the main paper.**
> >We have fixed Figure 2 based on your suggestion. Our revised Figure is included in the One-page PDF.
>
> **Q5: The motivation part deserves to be moved to the main paper.**
> >We are glad to hear you acknowledge our motivation, and we have moved the motivational example to our main paper.
>
> **Q6: Add text prompt conditions to MIME for your proposed task for a fair comparison.**
> >We implement your suggested method by concatenating CLIP text encoder features with the latent features at the transformer layer of MIME's architecture. Notably, by utilizing text prompts, MIME exhibits marginal improvements over the original results. Nevertheless, the outcome suggests that a latent strategy to incorporate text prompts upon existing works may be insufficient to solve the proposed problem effectively. We report the results in the following table.
>
> |||PRO-teXt|		||	HUMANISE|	|
> |-----------|---------------|--------------|-----------------|-----------------|-----------------|-----------------|
> |Baseline|	CD|	EMD	|F1|	CD	|EMD	|F1|
> |MIME|	2.0493	|1.3832	|0.0990	|5.4259	|2.0837	|0.0628|
> |MIME with text|	1.8424	|1.2865	|0.1032	|4.7035	|1.8201|	0.0849|
> |LSDM (ours)|**0.5365**	|**0.5906**	|**0.5160**	|**0.7379**	|**0.7505**	|**0.4395**|
>
> **Q7: Is the target object necessary to be the $M+1$ object?**
> >Certainly not. Our proposed method allows any object to be modified. If we want to change a specific object in the scene, we only need to rearrange the occurrence of other objects so that the target object is in the last order, and then execute the conditional generation.
>
> **Q8: What is the meaning of off-the-shelf in L171?**
> >Off-the-shelf means we utilize conventional architecture without modifying it. In this context, we leverage the standard implementation of a transformer encoder.
>
> **Q9: The output of the text encoder is 1D. Do you apply pooling here?**
> >No, we do not apply pooling. Instead, we utilize the features from the End-Of-Text (EOT) token, resulting in a 1D representation of the text prompt.
>
> **Q10: It seems that the authors do not properly discuss the limitation and the broader impact of their work.**
> >The limitation of our method is that the theoretical findings have an assumption constrained to uniform data like point clouds. The predicted guiding points are not always aligned with the target object, which is indicated in the results of Tables 1 and 4 of the main paper. Furthermore, the editing results show necessary improvements in future works. The broader impact of our paper lies in the potential applications of VR, animation, and metaverse. We have included these details in the final version of our paper. Thank you for your comments.
>
> References:
>
> [1] Ye et al. Scene synthesis from human motion. In SIGGRAPH Asia 2022.
>
> [2] Yi et al. MIME: Human-Aware 3D Scene Generation. In CVPR 2023

---

> > ### Author Response · Authors · 2023-08-15
> > **Looking forward to your reply**
> >
> > Dear Reviewer uwsL,
> >
> > Thanks for your hard work in the reviewing process! Please let us know if you have any further questions before the end of the author-reviewer discussion phase.
> >
> > Thanks, Authors.

---

> > ### Comment · Reviewer_uwsL · 2023-08-21
> > **Response to rebuttal**
> >
> > Thanks for the detailed rebuttal and the revision of the manuscript. The revision makes the presentation more clear now. I would like to raise my score to WA.

---

> > > ### Author Response · Authors · 2023-08-22
> > > **Thanks for your reconsideration**
> > >
> > > Dear Reviewer **uwsL**,
> > >
> > > Thank you for your reconsideration! Best regards, Authors.

---

### Author Rebuttal · Authors · 2023-08-07

**General Response**

Dear ACs and Reviewers,

Thanks for your valuable reviews and insightful comments, which have helped us improve our paper. During the initial reviews, Reviewers **uswL**, **aT28**, **c9eB** were inclined toward acceptance. We are glad that our proposed language-driven scene synthesis task "is novel" (Reviewer **RVaZ**), "is an interesting direction" (Reviewer **uswL**), can "alleviate the control burden from the designer's side" (Reviewer **aT28**), and "holds great promise and significance in bridging the gap between research and real-world applications" (Reviewer **c9eB**). We are also encouraged that our proposed guiding points network "is a cool contribution" (Reviewer **YsUe**) and "the experiments show the effectiveness of the proposed pipeline and the learning objective" (Reviewer **aT28**).

The common concern raised by Reviewers is the significance of Section 3.2 (Reviewer **uswL**, **aT28**, **c9eB**). We have explained that section 3.2 is important to our paper for two reasons. First, Remark 1.2 establishes Eq. (7), which indicates our paper's central assumption that guiding points $\tilde{S}$ explicitly contribute to the denoising process; thus, $\tilde{S}$ can be learned concurrently with the denoising process. Furthermore, $\tilde{S}$ of Eq. (7) *connects* theory with our network architecture. In the Motivation of the Appendix, we show that by applying transformation matrices to scene entities, we can predict the centroid of target object ($\tilde{S}$ of Eq. 7), leading to the network design in Section 3.3. Reviewer **YsUe** has questioned the working and design choice of our network architecture, we believe that the answer to this question is rooted in Remark 1.2, underscoring that Remark 1.2 is indeed significant. Second, in Section 3.2, we also establish Corollary 2.1, in which we have implied another intuitive observation in L152 that a smaller MSE between the predicted guiding points $\tilde{S}$ and $\mu_0$ corresponds to a more accurate estimation. This implication serves as a measurement method, and therefore, from the result of Table 4, we can conclude that our guiding points were meaningful and well-aligned with the centroids of target objects.

While we believe Section 3.2 is significant, we do take feedback from reviewers. We agree that Section 3.2 seems to use too much space (Reviewer **uswL**), and Proposition 2 and Corollary 2.1 may be more suitable to relocate them to the supplementary materials (Reviewer **c9eB**). We have shortened the interpretation of Remark 1.2 and moved Proposition 2 and Corollary 2.1 to the Appendix. We also revised some notations based on the suggestion of Reviewer **aT28** to avoid confusion and potential misunderstandings.

Finally, we have included more experiments, including MIME with text (suggested by Reviewer **uswL**), multi-conditional diffusion model (MCDM) (suggested by Reviewer **c9eB**, **YsUe**), and two-stage LSDM (suggested by Reviewer **YsUe**). The outcomes confirm that our proposed methods significantly outperform other comparative methods. We additionally include a study of the impact of the number of human pose frames on our LSDM in the One-page PDF. This study indicates that utilizing more human pose frames does not lead to better performance, thus, verifying that one human pose frame is enough for our method.

We are looking forward to responding to any further questions you have on our submission.

**Summary of Revision**

Integrating the suggestions and feedback from all reviewers, besides fixing typos and notations, we have made the following important updates in the revision.

   - We have included a Limitation section to describe the limitation of our method, including uniform assumptions of the theoretical findings and failure cases of predicting guiding points.

   - We have added a Broader Impact section to discuss the potential applications of our problem.

   - We have shortened the interpretation of Remark 1.2 and moved Proposition 2 and Corollary 2.1 to the Appendix.

   - We have updated the notations of Remark 1.2 to avoid confusion and potential misunderstandings.

   - We have moved the motivational example in Section 3 of the Appendix to the beginning of Section 3.3 of the main paper.

   - We have revised Figure 2 to describe the conducted operations fit better with the explanation in the text.

   - We have revised Section 4 in the Appendix, including more step-by-step explanations of our network architecture. Table 2 in the Appendix is also appended with a column to describe each component's functionality. A clear linkage is also ensured between the introduced explanations and Table 2.

   - We have added and discussed experiments with MIME with text, MCDM, and Two-stage LSDM in Table 1.

   - We have included a study on the impact of the number of human pose frames on scene synthesis results in Section 7 of the Appendix.

---

> ### Author Response · Authors · 2023-08-20
> **A friendly reminder**
>
> Dear Reviewers **uwsL**, **aT28**, **RVaZ**,
>
> We sincerely appreciate the time and effort throughout the reviewing process of our submission. As the author-reviewer discussion is due in about 1 day, please let us know if you have further questions about our submission.
>
> Once again, thank you in advance, and we look forward to your feedback.
>
> Best regards,
> Authors.

---

### Decision · Program_Chairs · 2023-09-21

**Decision:**

Accept (poster)

**Comment:**

This submission received generally positive reviews from the reviewers. Reviewer RVaZ was concerned about the motivation and evaluation.  The authors provided a response, but the reviewer did not engage in a discussion after a few reminders. The AC has read the submission, reviews, and responses, and found that the concerns are mostly addressed.  Therefore, the AC recommends acceptance.  The authors are encouraged to incorporate the feedback in the camera-ready version.